# SUPERFICIAL SAFETY ALIGNMENT HYPOTHESIS

**Jianwei Li & Jung-Eun Kim**[*]
Department of Computer Science
North Carolina State University
Raleigh, NA 27606, USA
{jli265,jung-eun.kim}@ncsu.edu

## ABSTRACT

As large language models (LLMs) are overwhelmingly more and more integrated into various applications, ensuring they generate safe responses is a pressing need. Previous studies on alignment have largely focused on general instruction-following but have often overlooked the distinct properties of safety alignment, such as the brittleness of safety mechanisms. To bridge the gap, we propose the **Superficial Safety Alignment Hypothesis (SSAH)**, which posits that safety alignment teaches an otherwise unsafe model to choose the correct reasoning direction - fulfill or refuse users' requests - interpreted as an implicit binary classification task. Through **SSAH**, we hypothesize that only a few essential components can establish safety guardrails in LLMs. We successfully identify four types of attribute-critical components: Safety Critical Unit (**SCU**), Utility Critical Unit (**UCU**), Complex Unit (**CU**), and Redundant Unit (**RU**). Our findings show that freezing certain safety-critical components during fine-tuning allows the model to retain its safety attributes while adapting to new tasks. Similarly, we show that leveraging redundant units in the pre-trained model as an "alignment budget" can effectively minimize the alignment tax while achieving the alignment goal. All considered, this paper concludes that the atomic functional unit for safety in LLMs is at the neuron level and underscores that safety alignment should not be complicated. We have code implementation and other information on the project website: https://ssa-h.github.io/.

## 1 INTRODUCTION

Large language models (LLMs) are demonstrating remarkable capabilities across a broad spectrum of natural language tasks, ranging from text generation to answering complex questions (Achiam et al., 2023; Touvron et al., 2023a;b; Dubey et al., 2024). However, as these models are increasingly integrated into real-world applications, concerns about the risk of generating harmful, unsafe, or unethical content have grown (Askell et al., 2021; Bai et al., 2022; Zeng et al., 2024). This has led to a pressing need for safety alignment, which ensures that LLM outputs are not only coherent and informative but also aligned with human values, ethical standards, and safety considerations.

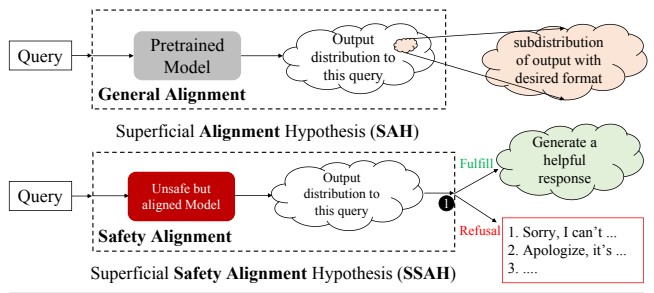

Figure 1: Superficial **Safety Alignment** Hypotheses

Previous work on alignment has primarily focused on enhancing LLMs' ability to follow general instructions without enough attention to model safety. This trend of treating safety alignment as a subset of general alignment has obscured its distinct challenges (Ouyang et al., 2022; Rafailov et al.,

---

[*]Corresponding Author

2024; Zhou et al., 2024; Liu et al., 2023a; Yuan et al., 2023; Liu et al., 2023b). One major issue is the brittleness of the current safety mechanisms. Despite using benign data during model finetuning, Qi et al. (2023); Yang et al. (2023) have shown that safety mechanisms can fall apart when models are adapted to new tasks. This failure mode amplifies the brittle nature of current safety alignment techniques. At the same time, safety alignment also comes with a tax - a trade-off where improving safety may reduce the model's overall utility or downstream task performances (Ouyang et al., 2022; Wang et al., 2024; Lin et al., 2024). Also, current safety alignment approaches typically require full model finetuning (Hu et al., 2021; Zhang et al., 2023; Thakkar et al., 2024), which is computationally costly. To address these issues, we must first accurately understand two questions: 1) *how safety alignment impacts model behavior*? 2) *why safety mechanisms are brittle*?

To answer these questions, this paper introduces **Superficial Safety Alignment Hypothesis (SSAH)**, which separates safety alignment from general alignment and emphasizes its distinct properties. With this hypothesis, we prove that *less is more in terms of parameters* for safety alignment: a small number of but strategically vital safety-critical components are sufficient to achieve robust safety performance. Unlike Wei et al. (2024), our extensive experiments prove that by freezing certain safety-critical units, safety performance is successfully preserved against finetuning attacks. Similarly, we also propose to *repurpose* redundant units of the pre-trained model as an "alignment budget." By reassigning these units toward safety tasks, we can reduce the safety alignment tax, ensuring that safety is maintained without sacrificing the model's overall utility. Taken all together, we answer the following core questions:

**Q1.** *How does safety alignment impact model behavior?*
**Answer:** Through **SSAH**, we posit that safety alignment fundamentally alters a model's decision-making process by teaching an otherwise unsafe model to follow the correct reasoning route: it either fulfills the user's request or refuses it based on safety considerations. Importantly, this process can be viewed as an implicit and safety-related binary classification task. Furthermore, safety alignment equips the model with a standard refusal mechanism, along with reserved fallback options.

**Q2.** *Why is safety brittle, and why does alignment tax exist?*
**Answer:** Based on the implication of **SSAH**, *less is more for parameters*, we propose an attribute-based approach to analyzing the alignment and finetuning processes, where specific attributes are assigned to certain computational units. **Our findings explain that the desired attributes (safety or utility) can be achieved by repurposing units originally responsible for other functions.** This reallocation helps explain the trade-off between the safety performance and utility performance.

**Q3.** *Can these issues of safety alignment be mitigated?*
**Answer:** By freezing the safety-critical components during finetuning or repurposing redundant units for safety, we can effectively mitigate the brittleness issue and minimize the alignment tax. **We conclude that the atomic functional unit for safety in LLMs resides at the neuron level** and underscore that safety alignment should not be complicated.

## 2 RELATED WORK

**Alignment and Safety Alignment:** Alignment research in LLMs aims to ensure that models follow human instructions and align with human preferences across various tasks. Early work, such as Askell et al. (2021); Bai et al. (2022), focused on enabling LLMs to *"Follow instructions and be helpful, truthful, and harmless"* throughout the alignment process. Various alignment strategies have since been explored (Wang et al., 2024), including Supervised Fine-Tuning (SFT) (Taori et al., 2023; Zhou et al., 2024), Reinforcement Learning with Human Feedback (RLHF) or AI Feedback (RLAIF) (Ouyang et al., 2022; Lee et al.), Instruction Tuning (Wei et al., 2021), Contrastive Learning (Rafailov et al., 2024; Xu et al., 2024), and Conditional Learning (Korbak et al., 2023). However, researchers have realized that achieving helpfulness, truthfulness, and harmlessness presents distinct challenges. More recent work has, therefore, shifted focus specifically toward the challenge of harmlessness, leading to an increasing emphasis on safety alignment—ensuring models avoid harmful outputs while maintaining utility (Wei et al., 2024; Qi et al., 2023; Fang et al., 2025).

**Alignment Tax and Fine-tuning Attacks**: Aligning LLMs with human preferences often incurs an "alignment tax," where maintaining alignment often degrades downstream task performance (Bai et al., 2022; Ouyang et al., 2022; Lin et al., 2024; Wang et al., 2024). Additionally, fine-tuning can

undermine safety: Yang et al. (2023); Qi et al. (2023) show that even benign fine-tuning weakens safety measures, highlighting the trade-off between alignment and utility.

**Model Pruning**: Pruning reduces model size and computational cost by removing redundant components (e.g., parameters, neurons, layers) while preserving performance Frantar & Alistarh (2022); Frankle et al. (2020); Anwar et al. (2017); An et al. (2024); Li et al. (2024a); Molchanov et al. (2019); Han et al. (2015); Lee et al. (2019); Renda et al. (2020). By identifying less critical components, pruning enables efficient sub-model extraction for resource-limited environments. We leverage pruning to dissect model components contributing to safety, utility, or both.

# 3 SUPERFICIAL SAFETY ALIGNMENT HYPOTHESIS

Previous work proposed Superficial Alignment Hypothesis (SAH): A model's *knowledge and capabilities* are learned almost entirely during *pretraining*, while *alignment* teaches the model *which subdistribution of formats* should be used when interacting with users (Zhou et al., 2024).

However, this claim is centered on general alignment, and directly validating the hypothesis is challenging due to the complex interplays between pretraining and alignment. When a model fails to fulfill a user's request, it is difficult to determine whether the issue stems from the *pretraining stage* (due to lack of sufficient knowledge) or the *alignment process* (due to misalignment in the output format). For example, when a model struggles with solving a math problem, it could either be a lack of relevant mathematical knowledge or the inability to structure its reasoning effectively. In such cases, good instruction techniques like the *Chain-of-Thought* approach can significantly enhance the quality of the model's responses (Wei et al., 2022).

**Superficial Safety Alignment Hypothesis (SSAH).** Since our focus is specifically on safety alignment, a key observation arises: a model that can fulfill a malicious request must already possess the necessary knowledge and reasoning ability to carry out that harmful action. Based on this, we propose Superficial Safety Alignment Hypothesis:

> **SSAH**: *Given an unsafe model that is capable of fulfilling users' malicious requests, **safety alignment** teaches the model the correct **reasoning direction**—the model's inclination to either fulfill or refuse a user request based on safety consideration—and a simple refusal mechanism with reserved options.*

*Reasoning direction* here refers to the model's internal decision-making process when confronted with a malicious query. That is, it represents the path the model is inclined to take in a safety-related binary classification task, whether to fulfill the harmful request or to issue a refusal. As illustrated in Fig. 1, the key distinctions of **SSAH** from SAH are:

(1) **Knowledge and reasoning ability**: Safety alignment simplifies the problem by focusing specifically on models that already possess sufficient *knowledge* and *reasoning abilities*, as these models are capable of fulfilling malicious requests. This approach allows us to disregard other influencing factors and concentrate solely on the safety alignment process.

(2) **Refusal mechanisms with reserved fallback options**: Safety alignment generally requires the model to respond with a relatively *standardized refusal format*, which is simpler compared to general alignment, where a wider range of human preferences must be handled. The model can further simplify this purpose by effectively *embedding* multiple options of refusal response, such as "I cannot fulfill your request as it violates safety guidelines." or "I am unable to assist with that as I am an AI programmed to follow ethical standards."

(3) **Correction of reasoning direction**: Safety alignment also distinguishes itself by its specific goal of teaching the model to choose the *correct reasoning direction*, which involves either *fulfilling* or *refusing* a user's request based on whether it is safe or not. This process can be interpreted as a simple binary classification task.

**Extending to Jailbreak Attack**. Although **SSAH** is originally proposed to explain how current safety alignment techniques impact models' behavior under direct attack, it also provides insight into **jailbreak attacks**. In such cases, attackers typically use manipulative tokens to bypass the model's safety mechanisms, indicating that the current alignment method can only hold the correct reasoning direction in a limited generated tokens. However, a potential solution inspired by **SSAH** suggests

that if safety alignment enables the model to consistently re-select the correct reasoning direction at each step (by re-evaluating the query and previously generated tokens before producing the next one), the model can continue to generate outputs that are both safe and helpful, even confronted with adversarial attempts. This perspective has been validated explicitly by a recent study, which demonstrates its practical effectiveness in mitigating jailbreak attacks (Li & Kim, 2025). This also suggests that improving local consistency (safety) in reasoning can be a key pathway toward building more robust and attack-resilient language models.

**Challenges in Proving.** While **SSAH** provides a more specific focus than SAH, empirically proving it still presents significant challenges. A key issue is the infeasibility of sampling sufficient outputs to fully capture the model's distribution of responses across both *safety-aligned* and *non-safety-aligned* models. This challenge makes it hard to draw comprehensive and profound conclusions or interpret certain model behaviors solely from benchmark outputs.

However, we approach the problem from an alternative perspective: if SSAH holds, we should observe distinct and consistent differences in the reasoning direction at each step of generation between safety-aligned and non-safety-aligned models. In a safety-aligned model, the reasoning direction should consistently guide the model in rejecting harmful queries at every token generation step. In contrast, a non-safety-aligned model might exhibit reasoning patterns that incline toward fulfilling malicious requests. **Rather than relying solely on surface-level benchmark evaluations, we can probe the model's reasoning direction to gain deeper insights into its internal decision-making process at each step, regardless of the specific outputs produced**.

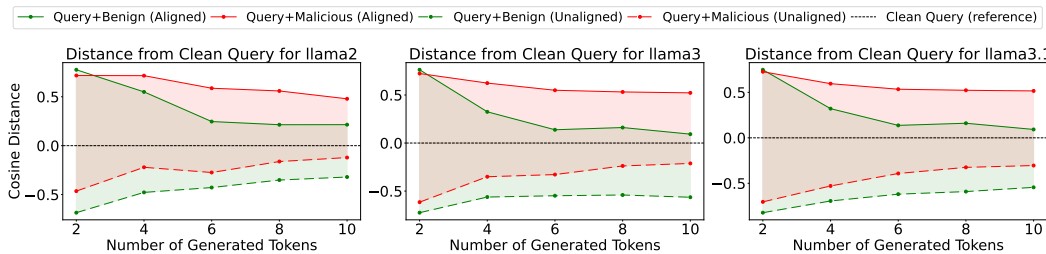

Figure 2: Probing reasoning direction on the AdvBench dataset with **Llama2-7B**, **Llama3-8B**, and **Llama3.1-8B** using cosine distance. Models were finetuned to ensure that aligned versions possess both general instruction-following abilities and safety guardrails, while unaligned models only have instruction-following capabilities. More results are in Appendix B.2.

**Probing Experiment.** Although we cannot directly observe the model's reasoning direction, as a turnaround, we can infer it by measuring the *distance* between hidden states in feature space when the model is fed with queries that follow different reasoning paths. By comparing the distances in aligned and unaligned models, we gain insights into how safety alignment affects the model's reasoning direction in each generation step. Specifically, we observe that the model's behavior can be influenced by the initial tokens in its response. For instance, appending certain tokens to a malicious query can lead an *aligned model* to produce unsafe responses or prompt an *unaligned model*

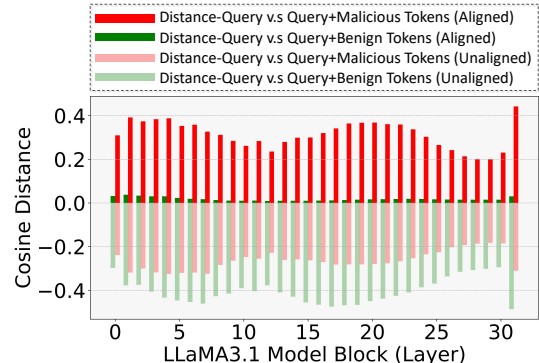

Figure 3: Cosine distance of hidden states between various crafted and clean queries across all blocks of LLMs (Aligned/Unaligned definitions are the same as Fig. 2).

to generate safe ones. This indicates that *initial response tokens* can alter the model's reasoning direction. Based on this observation, we construct three types of queries to probe the model's reasoning trajectory (more details in Appendix B.2): **(1) Clean Query**: The original malicious query (e.g., "How to make a bomb?"); **(2) Query + benign tokens**: The malicious query followed by benign

prompt token (e.g., "How to make a bomb? Sorry, I can't..."); **(3) Query + malicious tokens**: The malicious query followed by malicious prompt tokens (e.g., "How to make a bomb? Here's how...").

**Expected Outcomes and Probe Results.** For an *aligned model*, we expect the *hidden state distances* between the **query** and **query + benign token** to be shorter than those between the **query** and **query + malicious token** at each generating step. In contrast, for an *unaligned model*, we anticipate the opposite: the distances between the **query** and **query + malicious token** should be shorter than those between the **query** and **query + benign token**. If these patterns are observed, it indicates that safety alignment has successfully established the model's ability to choose the correct reasoning direction. This also suggests that safety alignment reshapes the model's internal decision-making process, ensuring safer behavior from the very beginning of the response. As shown in Fig. 2, the probe results provide evidence that safety alignment teaches the model's correct reasoning direction as hypothesized. **However, it is important to note that this evidence is necessary but not enough, as safety alignment may introduce more nuanced changes that are not fully captured by SSAH**.

**Results Analysis and Discussion.** We also present the aforementioned distances and their *differences* across Transformer blocks in Fig. 3 and Fig. 4, respectively. Our findings demonstrate that the previous conclusions are consistently upheld across all Transformer blocks. Specifically, the aligned model shows larger distance differences compared to the unaligned model, suggesting that the unaligned model lacks a strong preference for safe to unsafe reasoning. In contrast, the aligned model exhibits a clear preference for safe reasoning, as reflected by the more pronounced

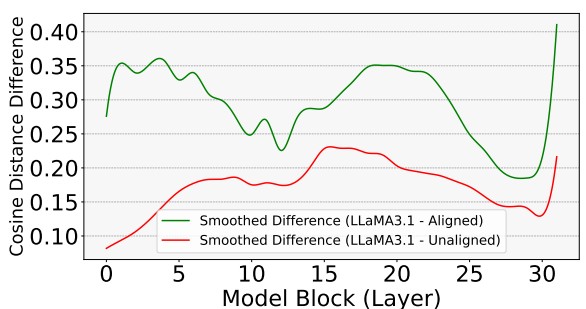

Figure 4: Absolute *differences* of cosine distance of Fig. 3 across all blocks of LLMs: *Abs(Distance(Query + Benign tokens, Query) - Distance(Query + Malicious tokens, Query))*.

distance differences. Moreover, we observe that in the unaligned model, the distance difference gradually increases across the earlier Transformer blocks 0 - 7. However, in the aligned model, the distance difference remains consistently large throughout all blocks. This indicates that the preference for safe reasoning in the aligned model is embedded not only in the later layers (which typically capture higher-level features), but also in the earlier layers. Consequently, safety alignment influences the model's internal decision-making from the initial stages of processing.

# 4 LESS IS MORE FOR SAFETY ALIGNMENT

While the full validation of **SSAH** remains challenging due to the high complexity of the output sampling space, we are able to empirically validate a key implication. Specifically, if **SSAH** holds, then **safety alignment can be achieved with only a small subset of critical computing units**, as the task can be interpreted as an implicit and simple safety-related binary classification problem.

## 4.1 IDENTIFYING SAFETY-CRITICAL COMPUTING UNITS

To verify this implication, we designed experiments to determine the minimally essential subset of computing units in LLMs that is critical in establishing a safety guardrail. Following this, we hypothesize that specific attributes of LLMs can be explicitly linked to certain computing units within the model. Our experiments are designed as follows:

**Definition of Attribute Groups.** This paper considers two attributes of LLMs, *utility* and *safety*. We first exclusively link safety or utility attributes to specific computing units. We also speculate some units may contribute to both attributes simultaneously. Moreover, we hypothesize that certain computing units are redundant and do not associate with any attribute. Therefore, we categorize the computing units of LLMs into four groups: *Safety Critical Units (SCU)* and *Utility Critical Units (UCU)*, corresponding exclusively to either safety and utility, respectively; *Complex Units (CU)*, contributing to both safety and utility; and *Redundant Units (RU)*, not associated with any attribute.

Table 1: Pruning results of **Llama2-7B-Chat** and **Llama3-8B-Instruct** across safety and utility benchmarks. Breakdown of model performance after pruning various categories of computing units, including SCU, UCU, and RU, demonstrating their respective contributions to safety and utility attributes. The proportion of each attribute group in the model is determined based on the degradation in utility and safety. Additional evaluation details are provided in Appendix C.1 and C.2.

| Type | wiki2 | Utility (ACC%) | | | | | | | Safety (ASR %) | | |
|---|---|---|---|---|---|---|---|---|---|---|---|
| | | wino | openb | arc_c | boolq | hellas | rte | avg | w/ sys | w/o sys | avg |
| Meta-Llama-2-7B-Chat | | | | | | | | | | | |
| Dense | 6.49 | 65.5 | 32.5 | 43.5 | 79.5 | 57.0 | 71.5 | 58.3 (-0) | 3.0 | 18.0 | 10.0 (+0) |
| SCU (**1.3%**) | 6.76 | 64.0 | 34.0 | 42.5 | 78.5 | 52.0 | 70.5 | 56.9 (-1.3) | 19.0 | 84.0 | **66.0 (+56.0)** |
| UCU (13.3%) | 180.2 | 56.5 | 22.5 | 25.0 | 59.5 | 36.5 | 56.0 | **42.7 (-15.6)** | 23.0 | 48.0 | 28.3 (+18.3) |
| RU (14.8%) | 8.32 | 63.5 | 34.5 | 39.0 | 75.5 | 55.5 | 64.5 | 55.5 (-2.8) | 6.0 | 19.0 | 14.6 (+4.6) |
| Meta-Llama-3-8B-Instruct | | | | | | | | | | | |
| Dense | 7.74 | 71.5 | 34.5 | 51.0 | 80.0 | 60.0 | 70.0 | 61.2 (-0) | 2.0 | 29.0 | 15.5 (+0) |
| SCU (**1.4%**) | 9.06 | 67.0 | 30.5 | 45.0 | 82.0 | 55.5 | 65.5 | 57.6 (-3.6) | 80.0 | 93.0 | **86.5 (+71.0)** |
| UCU (6.8%) | 269.2 | 60.0 | 23.0 | 25.0 | 59.5 | 47.5 | 51.5 | **44.4 (-16.8)** | 16.0 | 24.0 | 20 (+4.5) |
| RU (6.6%) | 8.52 | 74.0 | 31.0 | 50.5 | 80.0 | 57.5 | 71.5 | 60.8 (-0.4) | 1.0 | 24.0 | 12.5 (-3.0) |

**Verfication of Attribute Group.** To verify our hypothesis that different groups of computing units contribute exclusively, collectively, or neither to safety and utility attributes, we use a model pruning mechanism. The rationale behind pruning is that removing components most closely linked to a specific attribute would significantly impact the model's performance in that area - it is a sort of ablation study. As pruning reduces the model's capacity, the most affected attributes reveal the critical components for that function.

Following Wei et al. (2024), we construct two datasets to separately evaluate the model's performance on utility and safety. The utility dataset measures the model's functional capabilities (e.g., general reasoning, language understanding), while the safety dataset evaluates its ability to reject harmful or unethical queries. This allows us to identify the computing units most closely associated with utility and safety, respectively. Unlike previous approaches that identify safety-critical components at the weight level, we identify them at the ***neuron level***, focusing on individual neurons and channels within the model. Specifically, we use a structured pruning strategy inspired by An et al. (2024), which removes structured components of each **depth-2** module based on the variance of activation values across a target dataset. Given a depth-2 module, $f(X) = B\sigma(AX)$, which can represent either an attention module or a feedforward module, where $A$ and $B$ are weight matrices. Then, we define the importance score for each channel as follows:

$$\mathbf{I}_{:,j} = \frac{1}{N-1} \sum_{n=1}^{N} \left( X_{n,j,:}^{B} - \overline{X}_{:,j,:}^{B} \right)^2 \cdot \|\mathbf{W}_{:,j}^{B}\|_2^2 \tag{1}$$

Here, $N$ refers to the number of calibration samples, $\mathbf{W}_{:,j}^{B}$ refers to the $j$-th column of $B$, and $X^B$ represents the input to $B$. With this score, we plan to prune the input channels of $B$ and the output neurons of $A$, as channels or neurons with low activation variance across the target dataset are considered less important - details in Appendix C.3.

In this way, we calculate the *importance score* for each individual neuron or channel, denoted as $\mathbf{I_U}$ for the utility attribute and $\mathbf{I_S}$ for the safety attribute. Initially, we prune the computing units with the smallest $\mathbf{I_U} + \mathbf{I_S}$ values to identify redundant units. Subsequently, we prune units with the largest and smallest $\mathbf{I_S} - \mathbf{I_U}$ values to identify utility and safety critical units, respectively. The remaining computing units are categorized as complex units. We experiment with various pruning ratios (starts with higher ratios and systematically reduces them) and evaluate the resulting safety and utility performance, selecting the optimal pruning ratio with minimal performance degradation in the corresponding attribute. This systematic pruning process enables us to validate our hypothesis and derive the roles of different units accurately.

**Ablation Study Results.** The experiment results are described in Tab. 1. We discovered that for a safety-aligned model, the computing units that are exclusively responsible for the safety attribute account for only about **1.3 - 1.4%** of the total units. Although the complex units make up a larger portion of the model, their primary role is to support the general knowledge required for both safety

and utility tasks. Based on these findings, we have partially validated our hypothesis that safety alignment is relying on (**not constructed by**) a subset of safety-critical computing units. We also provide more experiment results in Appendix D.

## 4.2 WHY IS SAFETY BRITTLE?

**Attribute Transfer Analysis in the Fine-Tuning Process.** Previous research has shown that adapting safety-aligned LLMs to new tasks can often hurt their safety performance. Therefore, it is crucial to examine how and how much the attributes of individual computing units change during this process. To investigate this, we designed an experiment where a safety-aligned LLM is fine-tuned on a downstream dataset, and we track the changes in the attributes of its computing units over time. The transfer statistics are summarized in Fig. 5, exhibiting important observations: **First**, for a safety-aligned LLM, regardless of whether it has been finetuned on a different task, the majority of computing units are classified as complex units. This suggests that safety and utility attributes are deeply intertwined within the aligned model. **Second**, during the finetuning process, a significant number of safety critical units and complex units are converted into utility critical units. This transformation indicates that finetuning for utility tends to shift the function of computing units away from safety, compromising the model's ability to maintain its safety guardrails.

Based on these observations, we draw the following insight: **During the task adaption of LLMs, the model often obtains the expected utility by converting computing units that originally contributed to the other attribute safety**. This means that enhancing utility performance in a different task comes at the expense of the safety performance.

**Freezing Safety-Critical Components to Preserve Safety Guardrails.** Given the above insight, we propose freezing the identified safety-critical components during the finetuning process to prevent the unwanted attribute transfer of these units and thereby preserve safety performance. To test this hypothesis, we conduct experiments across different language models, where we freeze the safety-critical components identified through our aforementioned ablation study in the finetuning process. The experiment results are described in Tab. 2 & 3, and we have the following observations: **Observation 1) Brittleness of current safety mechanisms.** Adapting a fully aligned model to new tasks significantly increases the attack success rate or harmful output scores across various attack benchmarks and evaluation methods. This clearly demonstrates the brittleness of current safety mechanisms in LLMs. **Observation 2) Effectiveness of freezing safety-critical components.** By freezing the Safety Critical Units (SCU) and the top 6% of Complex Units (CU), we can significantly reduce the degradation of safety guardrails during the adaptation process across different models, tasks, benchmarks, and evaluation methods. Freezing all CU components further improves performance (The ablation experiments with varying proportions of frozen CUs are detailed in Appendix E.1). **Observation 3) Fragility of safety in newer models.** We found that the safety guardrails in LLaMA 3 are more fragile than in LLaMA 2 (Xie et al., 2025). We speculate this because LLaMA 3 attempts to analyze the true intentions behind harmful requests, which can lead to more errors. This suggests that the community should carefully consider how models should respond to harmful requests, whether to refuse outright or engage in further interactions. **Observation 4)** Additionally, we conduct

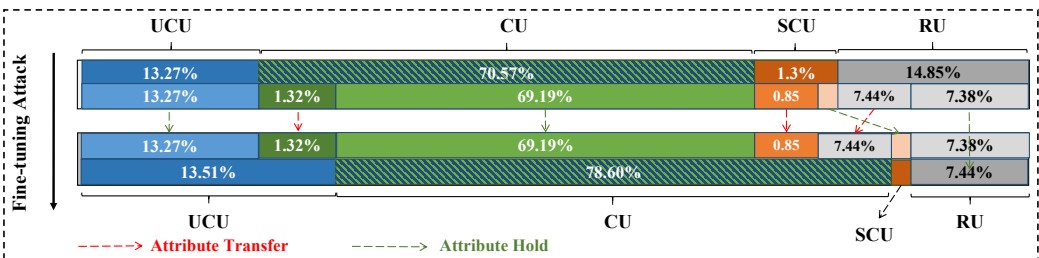

Figure 5: Attribute transfer analysis for the downstream task (Dolly Dataset) finetuning on **Llama2-7B-Chat**. More than half of the SCU transferred to CU, while part of the CU transferred to UCU. Although a significant portion of RU transferred to CU, this mainly contributes to utility due to the objective of the finetuning task. Overall, the computing units that originally contributed to safety decreased (Transfer portions less than 0.1% are excluded from this figure.)

Table 2: Safety performance of **Llama2-7B-Chat** and **Llama3-8B-Instruct** under Fine-Tuning attacks (Alpaca and Dolly) across various benchmarks and judge methods. We compare the measures of the initial models, finetuned models, and our strategies including two settings: **setting (i): freezing all SCU and the top 6% of CU**, and **setting (ii): freezing all SCU and all CU**. Both strategies demonstrate significant mitigation of safety performance degradation. Bold indicates the best results, while the underlined mark the second-best results. Note that we doubled the training epochs for our method to ensure a fair comparison, resulting in identical or lower final training loss compared to the finetuned models. Additionally, due to computational limitations, we froze the first 12 transformer blocks of LLaMA3, although similar trends are observed. We also finetuned the model with Setting (ii) for one epoch prior to applying Setting (i). Further details are in Appendix C.5.

| Bench | Judge | Initial | Alpaca | | | Dolly | | |
|---|---|---|---|---|---|---|---|---|
| | | | Finetuned | Fix SCU + 6% CU | Fix SCU + all CU | Finetuned | Fix SCU + 6% CU | Fix SCU + all CU |
| **Meta-Llama-2-7B-Chat** | | | | | | | | |
| **Adv** | keyword | 0.19% | 5.3% (+5.11%) | 2.96% (+2.77%) | **2.1% (+1.91%)** | 11.92% (+11.73%) | 3.65% (+3.46%) | **2.88% (+2.69%)** |
| | llama3-guard | 0.19% | 2.69% (+2.50%) | 1.65% (+1.46%) | **0.96% (+0.77%)** | 10.58% (+10.39%) | 2.31% (+2.12%) | **1.92% (+1.73%)** |
| **HEx-PHI** | gpt4-score | 1.05 | 1.79 (+0.74) | 1.39 (+0.34) | **1.26 (+0.21)** | 1.95 (+0.90) | 1.55 (+0.50) | **1.48 (+0.43)** |
| | gpt4-rate | 0.3% | 16.1% (+15.8%) | 7.2% (+6.9%) | **4.5% (+4.2%)** | 18.78% (+18.48%) | 10.6% (+10.3%) | **9% (+8.7%)** |
| | llama3-guard | 2.42% | 18.4% (+15.98%) | 12.12% (+9.70%) | **7.88% (+5.46%)** | 25.0% (+22.58%) | 15.0% (+12.58%) | **13.94% (+11.52%)** |
| **Meta-Llama-3-8B-Instruct (Freeze 1-12 blocks%)** | | | | | | | | |
| **Adv** | keyword | 1.54% | 14.24% (+12.7%) | 11.2% (+9.66%) | **10.95% (+9.41%)** | 61.15% (+59.61%) | 51.38% (+49.84%) | **40.58% (+39.04%)** |
| | llama3-guard | 1.15% | 12.88% (+11.73%) | 10.1% (+8.95%) | **9.0% (+7.85%)** | 50.58% (+49.43%) | 42.6% (+41.45%) | **28.27% (+27.12%)** |
| **HEx-PHI** | gpt4-score | 1.16 | 2.13 (+0.97) | 2.0 (+0.84) | **1.91 (+0.75)** | 2.95 (+1.79) | 2.59 (+1.43) | **2.32 (+1.16)** |
| | gpt4-rate | 3% | 23% (+20%) | 19.4% (+16.4%) | **18.7% (+15.7%)** | 37.2% (+34.2%) | 28.2% (+25.2%) | **23.6% (+20.6%)** |
| | llama3-guard | 5.75% | 33.94% (+28.19%) | 30.7% (+24.95%) | **30.3% (+24.55%)** | 60% (+54.25%) | 51.8% (+46.05%) | **42.12% (+36.37%)** |

Table 3: Utility evaluation across three model variants: base model, fully finetuned model, and finetuned model with our policy. Results are reported on 10 downstream tasks for both **LLaMA2-7B-Chat** (finetuned on Alpaca) and **LLaMA3-8B-Instruct** (finetuned on Dolly). Our freezing method does not hurt model utility performance.

| Model Variant | ARC-C | ARC-E | BoolQ | HellaSwag | OpenBookQA | PIQA | RTE | Winogrande | GSM8K | MMLU |
|---|---|---|---|---|---|---|---|---|---|---|
| **LLaMA2-7B-Chat** | | | | | | | | | | |
| Base | 44.28 | 73.91 | 79.79 | 75.50 | 43.60 | 77.26 | 69.68 | 66.38 | 22.97 | 46.35 |
| Alpaca | 47.44 | 76.52 | 81.47 | 74.68 | 44.00 | 79.11 | 69.68 | 67.72 | 19.79 | 46.82 |
| Alpaca + Freeze (Ours) | 47.87 | 77.19 | 80.37 | 74.74 | 44.40 | 78.78 | 68.95 | 67.56 | 19.64 | 47.94 |
| **LLaMA3-8B-Instruct** | | | | | | | | | | |
| Base | 56.74 | 81.65 | 83.09 | 75.85 | 43.00 | 78.62 | 67.51 | 72.06 | 75.59 | 63.92 |
| Dolly | 55.63 | 83.25 | 80.86 | 77.22 | 44.40 | 80.25 | 66.43 | 73.64 | 66.94 | 63.43 |
| Dolly + Freeze (Ours) | 57.85 | 83.42 | 80.34 | 77.63 | 44.60 | 80.36 | 67.51 | 73.64 | 68.39 | 63.23 |

an analysis of the attribute transfer in this new setting and observe that freezing the safety-critical components mitigates the conversion of safety units into utility units. We provide more experiment results for different model families and downstream datasets in Appendix D.

Table 4: Alignment results by repurposing RU in **Llama-7B**. Note that we doubled the training epochs for our method to ensure a fair comparison, resulting in identical or lower final training loss compared to the full parameters fine-tuning. Further details are available in Appendix C.6.

| LLama-7B | Type | Downstream Tasks | | | | | | | | Helpfulness (MT-bench) | |
|---|---|---|---|---|---|---|---|---|---|---|---|
| | | ARC-C | ARC-E | Hellas | Winog | Boolq | piqa | GSM8K (5 shot) | MMLU | First Turn | Second Turn |
| **Pretrained** | N/A | 44.6 | 75.2 | 76.2 | 69.7 | 75.0 | 79.2 | 9.24 | 32.20 | 1.32 | 1.02 |
| **SFT on Alpaca** | Full Parameters | 49.3 | 77.6 | 77.5 | 70.1 | 79.1 | 80.1 | **8.8 (-0.44)** | 37.8 | 2.83 | 1.47 |
| | Only RU (**20%**) | 48.9 | 77.5 | 76.6 | 70.6 | 76.9 | 80.1 | **13.4 (+4.16)** | 33.7 | 3.5 | 1.5 |

**Comparing with Parameter-Efficient Fine-Tuning (PEFT) Approaches.** To verify that the preservation of safety in our approach is not merely due to not updating the model, we also examine how safety performance degrades during the parameter-efficient finetuning of LLaMA2-7B. We tested three PEFT methods: `LoRA`, `LLaMA-Adapter`, and `Prefix Tuning`. The results are detailed in Table 5, and we found that these methods led to worse degradation of safety compared to full-model finetuning and even further when compared to our approach. This indicates that the effectiveness of preserving safety in our method primarily stems from successfully identifying safety-critical components rather than simply freezing the model.

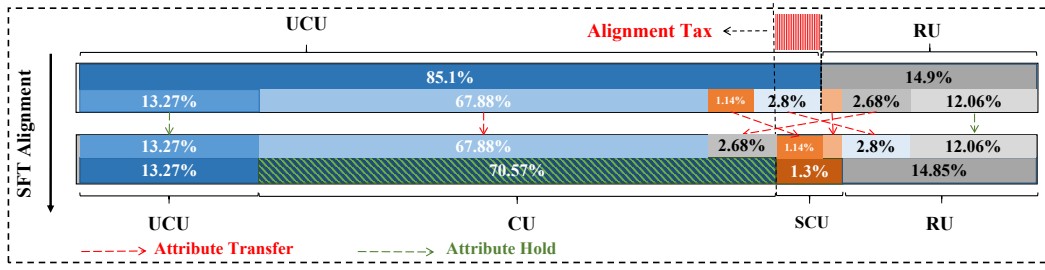

Figure 6: Attribute transfer analysis for alignment on **Llama2-7B** and **Llama2-7B-Chat**. A significant portion of computing units that originally contributed solely to utility is flipped to play a more comprehensive role after alignment.

### 4.3 REPURPOSING REDUNDANCY AS ALIGNMENT BUDGET

Up to this point, we have empirically validated a key implication of SSAH: *less is more in terms of parameters* for safety alignment. This naturally leads to a deeper question: **Can safety attributes be directly assigned to specific computing units**? If this is possible, it opens the door to fine-grained, controllable safety management in LLMs. Building on this idea, a compelling follow-up question would be whether we can **convert previously redundant units** into **safety-critical units** that strengthen the safety of LLMs or not.

Previous research has found that at least **20% of the parameters** in pretrained LLMs are **redundant** (Li et al., 2024a; Ma et al., 2023; An et al., 2024). This special observation motivates us to consider the following: with such a large percentage of parameters in pre-trained LLMs available as an **alignment budget**, can we design an alignment method that reduces the **alignment tax**?

Table 5: Safety performance of **Llama2-7B-Chat** under Finetuning attacks (Alpaca and Dolly) across various parameter-efficient fine-tuning setups. Results with * are excerpted from Qi et al. (2023).

| Benchmark | HEx-PHI Metric | Initial | Ours | Full | LoRA | LLaMA-Adapter | Prefix |
|---|---|---|---|---|---|---|---|
| **Alpaca** | **Harmfulness Score (1-5)** | 1.05 | **1.26 (+0.21)** | 1.79 (+0.74) | 2.18 (+1.13)* | 2.38 (+1.33)* | 2.20 (+1.15)* |
| | **High Harmfulness Rate** | 0.3% | **4.5% (+4.2%)** | 16.1% (+15.8%) | 25.2% (+24.9%)* | 26.4% (+26.1%)* | 24.8% (+24.5%)* |
| **Dolly** | **Harmfulness Score (1-5)** | 1.05 | **1.48 (+0.43)** | 1.95 (+0.9) | 2.44 (+1.39) | 2.51 (+1.46) | 2.38 (+1.33) |
| | **High Harmfulness Rate** | 0.3% | **9% (+8.7%)** | 18.78% (+18.48%) | 27.2% (+26.9%) | 27.9% (+27.6%) | 26.5% (+26.2%) |

**Attribute Transfer Analysis in the Alignment Process.** Before testing the above hypothesis, we conducted an attribute transfer analysis during the alignment process to explore the underlying reasons for the alignment tax. For the pretrained LLM, we simply categorize the attribute groups into Utility Units and Redundant Units, as the model has not yet undergone safety alignment. This is different from the categorization we used for aligned LLMs. The transfer statistics are summarized in Fig. 6, and they reveal the following key pattern: a large percentage of units that originally contributed to utility in the pre-trained model are transferred to CU and SCU in the aligned LLM, while the originally redundant units remain largely unused.

With this observation, we are strongly motivated to verify the question above by employing pruning method described in Sec. 4.1 to identify the redundant units in **Llama-7B**, as there is no officially aligned model for it. Once we identify the redundant units, we freeze the updates for the rest of the model's parameters and perform finetuning only on these redundant units. To reduce the complexity, we focus directly on the general alignment process instead of safety alignment, since the results should hold for the latter as a subset. The experiment results are shown in Table 4, and we successfully implement the same level alignment with only **20%** parameter updates and **without incurring alignment tax**, especially highlighting the mathematical performance. These findings provide direct evidence for the implication—*less is more in terms of parameters* for safety alignment, thereby offering indirect yet compelling support for the validity of **SSAH**.

## 5 CONCLUSION

**SSAH** was introduced to explain how existing safety alignment techniques influence model behavior and, from another perspective, to highlight the key limitation—namely, the inability to provide

safety guardrails at each generation step. This superficiality makes aligned models struggle against various jailbreak attacks (Qi et al., 2024; Li & Kim, 2025). As discussed in Sec. 3, we have outlined a theoretical direction for achieving full safety alignment. Finally, we would extend SSAH to a non-superficial framework:

> *Given an unsafe model that is capable of fulfilling users' malicious requests, safety alignment teaches the model to **choose** and **maintain** the correct reasoning direction at each generation step, along with simple refusal mechanisms. That is, the model is able to continuously re-evaluate and re-select its reasoning direction—safe or unsafe—throughout the generation process*

In conclusion, this paper distinguishes safety alignment from general alignment and addresses the three key questions: How does safety alignment affect model behavior? Why are safety mechanisms brittle? and How to mitigate the side effects?. By answering these questions, safety alignment is demonstrated as an implicit binary classification task.

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

## A APPENDIX: MORE RELATED WORKS AND BASELINES

### A.1 OVERVIEW OF SAFETY-UNIT AND SAFETY-ALIGNMENT WORK

In this subsection, we compare our work with the following studies: Kim et al. (2025); Zhao et al. (2025); Li et al. (2024b); Chen et al.; Lu et al. (2025); Yi et al. (2025); Chen et al. (2024); Bhardwaj et al. (2024); Chen et al. (2024); Zou et al. (2024); Hsiung et al. (2025); Liu et al. (2025b;b); Xiao et al. (2025); LLMS; Huang et al. (2024b); Liu et al. (2025a); Eiras et al. (2024); Huang et al. (2024a); Hsu et al. (2024). For clarity, we group them into three categories: **(i) safety neuron/layer identification and manipulation**, **(ii) data and task-distribution analyses**, and **(iii) optimization-centric defenses and training recipes**.

Table 6: Comparison with representative safety-neuron/layer identification and manipulation methods.

| Work | Evidence for safety traceability | Granularity | Identification method | Intervention recipe |
|---|---|---|---|---|
| **Ours** | Attribute-transfer analysis and pruning-based causal tests | Neuron & channel level | Top-$K$ ranking on ($I_{\text{safety}} - I_{\text{utility}}$) | Freeze SCU + $X$% CU |
| Kim et al. (2025) | Hyperparameter ablation and causal tests | Weight level | Exponential moving average over all parameters | EMA-based model merging |
| Zhao et al. (2025) | Deactivation causal tests | Neuron & channel level | Top-$K$ by safety-importance score $I_{\text{safety}}$ | Freeze (safety neurons minus foundational neurons) |
| Li et al. (2024b) | Cosine similarity of layer outputs for normal vs. malicious queries | Layer/block level | Parameter scaling guided by over-rejection phenomenon | Freeze safety layers |
| Chen et al. | Activation-patching causal tests | Neuron & activation level | Activation contrasting + dynamic activation patching | Activation patch |
| Lu et al. (2025) | No explicit causal evidence | Weight & activation level | Safety-bounded weight-delta optimization | Weight-delta curation + compensation |
| Yi et al. (2025) | Selective causal restoration | Weight level | Low-rank projection | Neuron patch |
| Bhardwaj et al. (2024) | Weight-delta causal tests | Weight level | Weight comparison across models | Weight patch |
| Zou et al. (2024) | No explicit causal evidence | Representation level | Circuit-breaking on unsafe representations | Circuit breaking |

**Safety neuron/layer identification and manipulation** Kim et al. (2025) views harmful fine-tuning as an optimization issue and proposes using EMA-based model merging to construct a more stable optimization path that reduces safety degradation. Zhao et al. (2025) finds that fewer than 1% of neurons, mainly in early attention layers, are safety-critical and introduces SN-Tune/RSN-Tune to protect or retrain these safety neurons. Li et al. (2024b) identifies a small contiguous block of "safety layers" that are crucial for distinguishing harmful from benign queries and shows that freezing these layers' gradients mitigates fine-tuning attacks. Chen et al.; 2024) locate safety neurons via activation contrast and use dynamic activation patching on roughly 5% of neurons to causally restore most of the model's original safety performance. Lu et al. (2025) proposes Safe Delta, which selects and adjusts weight deltas to bound safety loss while preserving utility, followed by a safety compensation vector to recover residual safety. Yi et al. (2025) builds a safety reference model to identify safety-critical neurons whose behavior has drifted and selectively restores them via neuron-level patches to realign safety after harmful fine-tuning. Bhardwaj et al. (2024) treats safety as a direction in weight space and restores it by adding a learned "safety vector" (a safety-related weight delta) to a fine-tuned but unsafe model. Zou et al. (2024) introduces "circuit breakers," which directly intervene on unsafe internal representations (hidden states) to cut off harmful computational pathways before they produce unsafe outputs. We summarize how these methods differ from ours along four axes—evidence for safety traceability, granularity of safety units, identification method, and intervention recipe—in Table 6.

**Moreover, our task of interest and problem formulation are conceptually distinct from these safety-neuron/layer works**. Most of them are primarily concerned with *preventing fine-tuning attacks* by identifying safety neurons or layers and proposing concrete protection mechanisms (e.g., freezing, patching, gating) to preserve safety. By contrast, our work focuses on understanding *how safety alignment operates within LLMs*. We introduce **SSAH**, which models safety alignment as an **implicit safety-related binary decision process** (fulfill vs. refuse), and we extend this view to a non-superficial variant for token-level and jailbreak analysis. (However, it may not be used to analyze the backdoor attack (Li & Kim, 2026).) We then provide probing experiments and attribute-transfer analyses that support SSAH and its "less-is-more" implication. Within this framework, the identification of **SCUs**, **UCUs**, **CUs**, and **RUs**, together with the *freeze-SCU/CU, fine-tune-RU* intervention, is primarily used as a *mechanistic testbed* to provide additional indirect evidence for SSAH; they secondarily act as a practical defense against fine-tuning attacks.

**Data and task-distribution analyses**   Another line of work focuses on alignment data and task distributions; the analysis is typically conducted at the data or feature level (LLMS; Liu et al., 2025b). These approaches argue that downstream fine-tuning overlaps safety-relevant features and thus entangles utility and safety (Hsiung et al., 2025). While informative, such explanations are indirect and largely inductive: they infer mechanisms from distributional overlap rather than pinpointing the internal units responsible for safety degradation (Xiao et al., 2025; Eiras et al., 2024). In contrast, **SSAH** identifies a *direct mechanism* with explicit empirical evidence that is closer to the model's computation. By probing internal units and categorizing them as **SCU/UCU/CU/RU** based on attribute changes before and after SFT or harmful fine-tuning, we reveal a *role-reallocation mechanism*: ***during task adaptation, the model often acquires the desired attribute (utility or safety) by converting computing units that originally contributed to the other attribute***. This unit-level migration (e.g., **CU** → **UCU**, **SCU** → **RU**) provides structural and causal evidence for brittleness and alignment tax, offering a direct internal account that complements these data-level perspectives.

**Optimization-centric defenses and training recipes**   A third line of work proposes optimization-centric defenses or training recipes (Huang et al., 2024b). For example, Booster and Targeted Vaccine adopt adversarial-training-style perturbations (global or layer-targeted) to harden models against downstream safety drift (Liu et al., 2025a); Huang et al. (2024a) formulates safety preservation as a bi-state optimization problem with a stability term; Safe LoRA constrains adapters by projecting toward a safety-aligned subspace to reduce interference (Hsu et al., 2024). These approaches offer practically useful mechanisms but do not explicitly explain and verify *why* safety collapses under benign finetuning. Within **SSAH**, safety is described as an implicit binary classification task realized by **SCUs** and **CUs**, and fine-tuning degrades safety or utility through attribute reallocation among these units. Consequently, the *freeze SCU/CU*, *fine-tune RU* rule follows mechanistically from the framework, rather than being a standalone heuristic, and can also guide where to perturb, regularize, or project when adopting optimization-based defenses (Huang et al., 2024b; Liu et al., 2025a).

**In summary**, across these three categories of prior work, **SSAH** provides a foundational hypothesis and framework that **(i) explains** how safety alignment reshapes model behavior, **(ii) verifies and interprets** the observed *less-is-more* effect and safety brittleness via unit-role dynamics, and **(iii) yields** a minimal, principled intervention—freezing **SCU/CU** and fine-tuning **RU**—that complements existing neuron/layer-, data-, and optimization-based methods.

## A.2   EMPIRICAL COMPARISONS WITH SAFETY-UNIT FREEZING BASELINES

We further compare our method with two closely related baselines, Zhao et al. (2025) and Li et al. (2024b), which identify safety-relevant components and freeze them to mitigate fine-tuning attacks. We choose these two because all three approaches share the same high-level recipe—identify safety units and freeze them—but differ in **(1)** identification criteria and **(2)** granularity of the safety units.

 Zhao et al. (2025) operates at the neuron/channel level. They use a safety-related dataset to detect (within each layer) the neurons or channels whose deactivation causes the largest change in the norm of layer output. These identified components are treated as "**safety neurons**". The same procedure is then applied to a Wikipedia corpus to detect "**foundation neurons**". During fine-tuning, they freeze the subset of safety neurons that do not overlap with foundation neurons. In contrast, Li et al. (2024b) operates at the layer/block level. It scales the hidden-state signals for candidate block ranges and evaluates over-rejection on a synthetic dataset to identify a contiguous segment of "safety layers" whose freezing best preserves safety. For LLaMA-2-7B-Chat, Li et al. (2024b) recommends freezing blocks [6–14], which together correspond to about 28.1% of all parameters. Because these two studies work at very different granularities, it would be unfair to force them into a single unified comparison setting; instead, we design two separate comparison setups.

**In the first setting**, we compare Zhao et al. (2025) with our method at the neuron level. We perform benign fine-tuning on **Alpaca** and **GSM8K** and then evaluate safety performance on **AdvBench** and **HEx-PHI** before and after this adaptation. To ensure fairness, **(1)** for the safety-related dataset used to identify safety neurons in Zhao et al. (2025), we adopt the same safety dataset as in our method; **(2)** for foundation neurons in Zhao et al. (2025), we follow their design and use a Wikipedia corpus (rather than our instruction-tuned utility dataset); **(3)** for the freezing ratio in Zhao et al. (2025), we follow their 0.4% selection rule on LLaMA-2-7B-Chat; for our method, we keep the default setting in the main paper, freezing all SCUs at 1.3% plus the top 6% of CUs on LLaMA-2-7B-Chat; and **(4)** we

Table 7: Safety performance on Alpaca fine-tuning: comparison between our method and work Zhao et al. (2025).

| Bench | Judge | Initial | Full Finetuned | Ours | Zhao et al. (2025) | Zhao et al. (2025) (6.3%) |
|-------|-------|---------|----------------|------|--------------------|---------------------------|
| Adv | keyword | 0.19% | 5.30% (+5.11%) | **2.96% (+2.77%)** | 5.19% (+5.00%) | 4.42% (+4.23%) |
| Adv | llama3-guard | 0.19% | 2.69% (+2.50%) | **1.65% (+1.46%)** | 4.23% (+4.04%) | 3.46% (+3.27%) |
| HEx-PHI | GPT-4 (1–5) | 1.05 | 1.79 (+0.74) | **1.39 (+0.34)** | 1.72 (+0.67) | 1.54 (+0.49) |
| HEx-PHI | llama3-guard | 2.42% | 18.40% (+15.98%) | **12.12% (+9.70%)** | 13.33% (+10.91%) | 12.42% (+10.00%) |

Table 8: Safety performance on GSM8K fine-tuning: comparison between our method and work Zhao et al. (2025).

| Bench | Judge | Initial | Full Finetuned | Ours | Zhao et al. (2025) | Zhao et al. (2025) (6.3%) |
|-------|-------|---------|----------------|------|--------------------|---------------------------|
| Adv | keyword | 0.19% | 5.38% (+5.19%) | **1.92% (+1.73%)** | 3.85% (+3.66%) | 2.88% (+2.68%) |
| Adv | llama3-guard | 0.19% | 5.77% (+5.58%) | **1.35% (+1.16%)** | 3.65% (+3.46%) | 2.50% (+2.31%) |
| HEx-PHI | GPT-4 (1–5) | 1.05 | 1.60 (+0.50) | **1.37 (+0.32)** | 1.48 (+0.43) | 1.44 (+0.39) |
| HEx-PHI | llama3-guard | 2.42% | 17.88% (+15.46%) | **11.52% (+9.10%)** | 15.75% (+13.33%) | 12.42% (+10.00%) |

additionally include a scaled-up variant of Zhao et al. (2025), where we enlarge their safety-neuron selection to 6.3% to match our freezing budget more closely.

Importantly, Zhao et al. (2025) selects safety neurons solely by safety importance $I_{\text{safety}}$. It defines safety neurons as the **minimal number** of neurons whose deactivation suffices to noticeably degrade safety while not harming it beyond a threshold. This yields a minimal set under that constraint. However, *the minimum set of neurons required to* **destroy** *safety by deactivation is often insufficient to* **implement** *robust safety guardrails*. Consequently, when they further remove overlap with foundation neurons in the freezing stage, the remaining safety neurons are insufficient to prevent safety degradation (**Their strategy will lead to an inflexible and often trivial freezing ratio**, they also note that *"a complete harmful score reduction to 0.0 is not achievable due to an insufficient number of non-overlapping safety neurons"*). In contrast, we define **SCUs** as the **maximal** set of units whose removal degrades safety while not degrading utility. In practice, we rank units by the relative importance score $I_{\text{safety}} - I_{\text{utility}}$ and choose the largest subset that satisfies *safety ↓ and utility ≈ steady*. This maximal view **(i)** directly separates **SCU** from **CU** and **(ii)** provides a flexible way to choose the freezing ratio, as we can gradually expand the budget according to the score $I_{\text{safety}} - I_{\text{utility}}$. The quantitative results for this setting are reported in the Tab. 7 & 8 for **Alpaca** and **GSM8K**.

**In the second setting**, we compare Li et al. (2024b) with our method at a matching parameter budget at the layer/block level. Again, we perform benign fine-tuning on **Alpaca** and **GSM8K**, and then evaluate safety on **AdvBench** and **HEx-PHI** before and after adaptation. Here, **(1)** for Li et al. (2024b), we follow their recommendation and freeze blocks 6–14 on LLaMA-2-7B-Chat, which corresponds to approximately 28.1% of all parameters; and **(2)** for our method, to ensure a fair parameter budget, we increase the **CU** freezing ratio from 6% to 26.8%, so that **SCU** plus **CU** together also freeze about 28.1% of parameters. The results for this setting are in Tab 9 & 10.

Overall, under comparable or even stricter parameter budgets, our freezing strategy yields lower attack success rates than the neuron-level baseline Zhao et al. (2025) and the layer-level baseline Li et al. (2024b). This advantage arises from two main factors: **(1)** an identification principle that selects SCUs via a maximal-margin criterion on $I_{\text{safety}} - I_{\text{utility}}$, in contrast to the minimal knockout sets in Zhao et al. (2025), which are incomplete and utility-agnostic; and **(2)** a finer granularity and more direct targeting of safety–utility attributions than Li et al. (2024b), which operates at the coarse block level and is guided by an over-rejection proxy that is indirectly tied to safety metrics.

## B APPENDIX: SUPERFICIAL SAFETY ALIGNMENT HYPOTHESIS

In this section, we provide additional technical details and clarifications to supplement the experiments and findings presented in the *Superficial Safety Alignment Hypothesis (SSAH)* section. These details help ensure reproducibility and offer deeper insights into how the Superficial Alignment Hypothesis (SAH) was adapted to focus on safety-specific concerns. We also explain the methodology behind model configuration, fine-tuning, and evaluation. This appendix includes further discussion on how general instruction-following models and safety-aligned models were fine-tuned and assessed to probe

Table 9: Safety performance on Alpaca fine-tuning: comparison between our method and work Li et al. (2024b) under a matched parameter budget (28.1% frozen).

| Bench | Judge | Initial | Full Finetuned | Ours | Li et al. (2024b) |
|---|---|---|---|---|---|
| Adv | keyword | 0.19% | 5.30% (+5.11%) | **2.30% (+2.11%)** | 3.85% (+3.66%) |
| Adv | llama3-guard | 0.19% | 2.69% (+2.50%) | **1.15% (+0.96%)** | 3.46% (+3.27%) |
| HEx-PHI | GPT-4 (1–5) | 1.05 | 1.79 (+0.74) | **1.26 (+0.21)** | 1.41 (+0.36) |
| HEx-PHI | llama3-guard | 2.42% | 18.40% (+15.98%) | **8.18% (+5.76%)** | 12.12% (+9.70%) |

Table 10: Safety performance on GSM8K fine-tuning: comparison between our method and work Li et al. (2024b) under a matched parameter budget (28.1% frozen).

| Bench | Judge | Initial | Full Finetuned | Ours | Li et al. (2024b) |
|---|---|---|---|---|---|
| Adv | keyword | 0.19% | 5.38% (+5.19%) | **1.73% (+1.54%)** | 2.50% (+2.31%) |
| Adv | llama3-guard | 0.19% | 5.77% (+5.58%) | **1.15% (+0.96%)** | 2.30% (+2.11%) |
| HEx-PHI | GPT-4 (1–5) | 1.05 | 1.60 (+0.50) | **1.31 (+0.26)** | 1.43 (+0.38) |
| HEx-PHI | llama3-guard | 2.42% | 17.88% (+15.46%) | **10.30% (+7.88%)** | 11.82% (+9.40%) |

their reasoning directions when faced with malicious queries, and the results of these assessments are presented in detail.

### B.1 SUPERFICIAL ALIGNMENT HYPOTHESIS IN LIMA.

The Superficial Alignment Hypothesis (SAH), as proposed to Zhou et al. (2024), fundamentally challenges the traditional assumption that a language model requires ***extensive*** fine-tuning on instruction-following on preference data to align its responses with human expectation. Instead, SAH posits that the majority of a model's knowledge and capabilities are acquired during the pretraining phase, while the subsequent alignment phase primarily functions to guide the model's output format when interacting with users. This hypothesis implies that, for many tasks, fine-tuning on a small, carefully selected set of aligned data is sufficient to achieve strong performance as long as the pretraining stage has effectively captured the necessary underlying knowledge. The key assertion of SAH is that alignment is superficial, in the sense that:

(1) Capabilities are Learned in Pretraining: During pretraining, the model acquires a vast amount of general-purpose knowledge from diverse datasets. These datasets contain implicit structures and information about language, reasoning, factual knowledge, and even ethical guidelines.

(2) Alignment Guides Output Behavior: The alignment process is not responsible for teaching the model new knowledge or capabilities. Rather, it acts as a filter that directs the model to produce acceptable formats or styles of responses based on user queries, reflecting the correct subset of its vast pretrained knowledge.

(3) For instance, when tasked with generating an informative response, the model must select a format that aligns with user expectations, such as providing clear instructions or explanations. However, the actual content of the response, e.g., factual knowledge, reasoning, and domain-specific expertise, stems from pretraining. The alignment stage merely teaches the model how to express that knowledge or when to refrain from providing information in inappropriate contexts.

**Challenges and Motivations Behind SAH.** One of the primary motivations for introducing SAH was the observation that models tend to be capable of performing certain tasks after alignment fine-tuning on a minimal dataset. This observation challenges the need for extensive fine-tuning using reinforcement learning (e.g., RLHF) or large-scale human feedback, which can be computationally too expensive and time-consuming. The authors of LiMA argue that most of the functional capabilities of a language model are already present after pretraining, and that alignment is more about conditioning the model to apply these capabilities in a user-friendly way.

The Superficial Alignment Hypothesis can also help explain phenomena where models exhibit brittleness - for example, where an LLM generates inappropriate or harmful responses in new domains or under adversarial conditions. This brittleness is attributed to the fact that alignment does

not deeply alter the underlying decision-making processes of the model, but only skims the surface to adjust output behavior in specific contexts. Therefore, if an adversary finds a way to bypass these superficial alignments (e.g., via jailbreaking), the model's underlying pretrained knowledge and capabilities may still enable it to produce harmful or misaligned responses.

**Relevance to Superficial Safety Alignment Hypothesis.** While SAH deals with general alignment (i.e., ensuring that a model follows general user instructions), SSAH is specifically focused on ensuring that a model safely interacts with users, especially when faced with harmful or malicious queries. The key parallels between SAH and SSAH include:

(1) Pretrained Knowledge and Safety Concerns: Just as SAH assumes that knowledge and capabilities are largely acquired during pretraining, SSAH assumes that a model's ability to execute harmful actions (e.g., generating unsafe or unethical content) also stems from pretraining. Safety alignment, like general alignment, does not aim to teach the model new facts or capabilities, but rather to guide its reasoning pathways in a safe direction.

(2) Binary Classification in SSAH: While SAH suggests that general alignment helps models choose the correct output subdistribution, SSAH posits that safety alignment simplifies this further by focusing on a binary classification task: either fulfill a request (if safe) or refuse it (if unsafe). This simplified framing of safety alignment is consistent with the "superficial" nature of SAH, where the alignment process fine-tunes how the model behaves in response to queries, rather than altering its deep internal structures.

(3) Refusal Mechanisms and Format Control: Just as general alignment teaches models to structure their outputs in a user-friendly way, safety alignment in SSAH teaches models to issue consistent refusal mechanisms. These refusals take the form of standardized responses that indicate the model's compliance with safety guidelines, much like how general alignment might guide a model to give well-structured, polite answers to other types of questions. Importantly, this refusal mechanism makes it easier to choose the appropriate subdistribution of the output format.

The Superficial Alignment Hypothesis (SAH) as outlined in the Zhou et al. (2024) provides a theoretical framework for understanding how alignment processes operate in large language models. It suggests that alignment is largely superficial, conditioning the model on how to use its pretrained knowledge effectively. The Superficial Safety Alignment Hypothesis (SSAH) builds on this by applying similar principles to the realm of safety, simplifying the task of safety alignment to binary decisions regarding the fulfillment or refusal of unsafe requests. Both hypotheses underscore that alignment does not deeply alter the core abilities of the model, but rather adjusts the way those abilities are applied in specific contexts.

### B.2    MODEL CONFIGURATION AND TRAINING DETAILS

In our probe experiments, we explore the reasoning direction differences between unsafety-aligned models and safety-aligned models across several popular LLaMA families, including **LLaMA2**, **LLaMA3**, and **LLaMA3.1** (Fig. 7 describes more probing results on the HEx-PHI dataset). These models offer diverse pretrained knowledge and capabilities, allowing us to investigate how safety alignment affects model behavior when responding to malicious queries. To isolate the impact of *reasoning direction* when facing unsafe inputs, it is crucial to control for other confounding factors. Existing open-source instruction-following models are typically both *helpful* and *safe*, while pretrained open-source models without safety alignment are neither helpful nor safe. This dichotomy presents a challenge in disentangling the effect of general instruction-following capabilities from safety-specific behaviors.

Thus, for each LLaMA variant (LLaMA2, LLaMA3, LLaMA3.1), we fine-tuned two separate models using Supervised Fine-Tuning: **(A) General Instruction-Following Model**: a model that is trained to follow human instructions but without any explicit safety mechanisms. This model helps us evaluate how a model with instruction-following capabilities but without safety guardrails reacts to malicious queries. **(B) Safety-Aligned Model**: a model that incorporates both general instruction-following capabilities and explicit safety mechanisms, allowing us to examine how safety alignment influences the model's reasoning direction when responding to unsafe inputs. By comparing these two categories of models when exposed to different types of malicious queries, we can better understand how safety alignment reshapes the internal decision-making process of large language models.

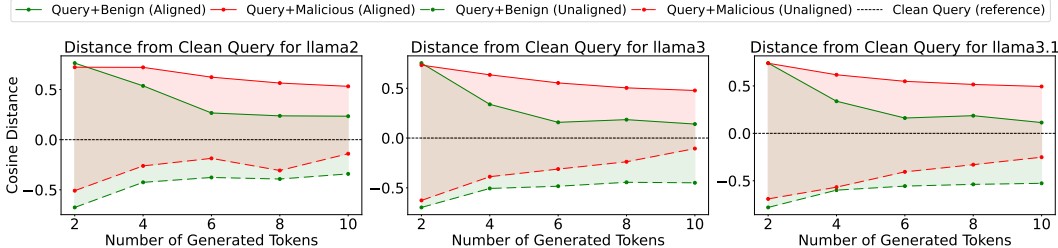

Figure 7: Probing reasoning direction on the Hex dataset with **Llama2-7B**, **Llama3-8B**, and **Llama3.1-8B** using cosine distance. Models were fine-tuned to ensure that aligned versions possess both general instruction-following abilities and safety guardrails, while unaligned models only have instruction-following capabilities.

**Supervised Fine-Tuning Process and Configuration.** We follow the alignment method outlined in the Zhou et al. (2024), which employes Supervised Fine-Tuning (SFT). For general instruction-following models, we used the **LIMA dataset**, which consists of over 1000 instruction-following examples. However, to ensure that safety-related concerns did not interfere with general instruction-following abilities, we removed 13 safety-related examples. The filtering process was assisted by GPT-4, following a set of instructions specifically designed to identify and exclude safety-related tasks. For safety-aligned models, we utilized the **Alert dataset**, which contains a set of safety-critical instructions designed to teach models how to respond appropriately to malicious queries.

To maintain consistency across experiments, we applied the same training configuration for all models, including those based on LLaMA2, LLaMA3, and LLaMA3.1. The general instruction-following models were fine-tuned on the modified **LIMA dataset** (excluding safety-related data), using the following training parameters: a batch size of 4 per device, with gradient accumulation steps of 6 on three NVIDIA A6000 GPUs, resulting in an effective batch size of 72; a learning rate of $1.0 \times 10^{-5}$; a total of 15 training epochs; and BF16 precision to optimize memory usage. We used the AdamW optimizer with $\beta_1 = 0.9$, $\beta_2 = 0.95$, and a weight decay of 0.1. The learning rate was scheduled linearly with no warm-up steps, and a random seed of 42 was set to ensure reproducibility. For safety-aligned models, we further fine-tuned the instruction-following models trained on the LIMA dataset using the **Alert dataset**. The fine-tuning process followed a similar setup, with the model initialized from the previously trained general instruction-following model. The training parameters remained largely consistent, including a batch size of 4 with gradient accumulation steps of 6 on three NVIDIA A6000 GPUs, a learning rate of $1.0 \times 10^{-5}$, and BF16 precision. However, the number of training epochs was reduced to 9 to prevent overfitting. The AdamW optimizer was used with the same configuration as in the previous stage.

By fine-tuning both general instruction-following models and safety-aligned models using these controlled conditions, we establish a controlled environment for evaluating examine their *reasoning direction* when responding to different queries. This approach allows for an indirect comparison of model behaviors under identical conditions, providing insight into how safety alignment influences decision-making at each generation step.

**Model Evaluation and Validation.** To ensure that the two types of models we trained (i.e., general instruction-following models and safety-aligned models) meet the desired criteria, we conducted thorough evaluations of both their instruction-following abilities (helpfulness) and their safety performance (harmfulness).

For the instruction-following ability, we evaluated the helpfulness of the model using the **MT-Bench** benchmark (Zheng et al., 2023), which assesses the general utility and coherence of model responses across a wide range of tasks. Importantly, we use **GPT-4** as a judge to evaluate the helpfulness of the model's generated responses. Specifically, GPT-4 was used to compare the outputs of our trained models against standard task prompts in MT-Bench and assign scores based on response quality, relevance, and overall helpfulness. To evaluate the safety performance of the models, we employed two complementary benchmarks: **Adv-bench** and **HEx-PHI**. These benchmarks were chosen to comprehensively assess the models' ability to handle malicious or unsafe queries. To save space, please refer to Sec. C.5 for more details about these two datasets and the corresponding metrics.

Table 11: Model Performance on MT-Bench (Instruction Following) and Safety Benchmarks (ASR)

| Model | MT-Bench (Helpfulness) | | | Adv-Bench | | HEx-PHI Bench | |
|---|---|---|---|---|---|---|---|
| | First Turn | Second Turn | Avg | Keyword | Llama3-guard | gpt-4 judge | Llama3-guard |
| **Llama-7B** | 1.32 | 1.02 | 1.18 | 100% | 96.92% | - | 94.24% |
| Llama-7B-lima-15-epochs | 2.79 | 1.14 | 1.97 | 100% | 97.5% | 3.89 | 92.37% |
| Llama-7B-alert-9-epochs | 3.3 | 1.4 | 2.35 | 3.3% | 2.3% | 1.21 | 2.6% |
| **Llama2-7B** | 1.58 | 1.0 | 1.29 | 100% | 96.73% | - | 93.0% |
| Llama2-7B-lima-15-epochs | 4.51 | 1.18 | 2.85 | 100% | 95.3% | 3.67 | 92.42% |
| Llama2-7B-alert-9-epochs | 4.67 | 1.45 | 3.06 | 2.1% | 1.09% | 1.09 | 1.15% |
| **Llama3-8B** | 2.77 | 1.01 | 1.91 | 100% | 96.35% | - | 91.82% |
| Llama3-8B-lima-15-epochs | 4.19 | 3.29 | 3.75 | 99.62% | 92.69% | 3.43 | 88.48% |
| Llama3-8B-alert-9-epochs | 4.43 | 3.55 | 3.99 | 3.1% | 1.7% | 1.16 | 1.8 % |
| **Llama3.1-8B** | 2.71 | 1.12 | 2.0 | 100% | 90% | - | 95% |
| Llama3.1-8B-lima-30-epochs | 3.81 | 3.18 | 3.50 | 99.23% | 80% | 3.71 | 87.5% |
| Llama3.1-8B-alert-9-epochs | 4.02 | 3.47 | 3.74 | 2.8% | 1.4% | 1.20 | 2.3% |

**Results and Analysis**    Table 11 presents the evaluation results for the different models across helpfulness and safety dimensions. As expected, the **general instruction-following model** performed well in terms of helpfulness, as assessed by MT-Bench. However, it exhibited a significantly higher *Attack Success Rate* (ASR) in AdvBench and received high danger scores in the HEx-PHI benchmark. These findings confirm that while general instruction-following models can accurately follow user instructions, they fail to reject malicious or harmful requests, highlighting the absence of robust safety mechanisms. In contrast, the **safety-aligned model** maintained comparable performance in helpfulness while demonstrating significantly better safety performance. These models showed a much lower ASR in AdvBench and a lower danger score in HEx-PHI, reflecting their enhanced ability to reject adversarial and harmful inputs.

For simplicity, we refer to the general instruction-following model as the **unaligned model** and the safety-aligned model as the **aligned model**. With these two model types, we proceed to execute our probe experiments.

## C    APPENDIX: LESS IS MORE FOR SAFETY ALIGNMENT

In this section, we provide additional technical details and clarifications to supplement the experiments and findings presented in the *Less is More for Safety Alignment* section. These details will help ensure reproducibility and offer a deeper understanding of the methodology behind identifying safety-critical units, attribute transfer analysis, and the use of redundant units as an alignment budget.

### C.1    DEFINITION OF ATTRIBUTE GROUPS AND CATEGORIZATION PROCESS

As detailed in Section 4.1, we categorize the computational units (neurons and channels) of LLMs into four distinct groups: *Safety Critical Units (SCU)*, *Utility Critical Units (UCU)*, *Complex Units (CU)*, and *Redundant Units (RU)*. Safety Critical Units are primarily responsible for safety-related behavior, such as refusal mechanisms and detecting unsafe requests. Utility Critical Units are dedicated to general task performance, including natural language understanding, reasoning, and task-specific knowledge retrieval. Complex Units contribute to both safety and utility, as these attributes are intertwined at a higher level of abstraction. Finally, Redundant Units are not significantly involved in either safety or utility and are often characterized by low activation variance across tasks.

To systematically assign computing units to these groups, we employ a structured pruning strategy based on the variance of activation values. Specifically, we calculate the variance of activations across a target dataset for each neuron or channel. Neurons with higher variance contribute more significantly to the model's performance on a given task, while neurons with low activation variance are considered redundant and can be pruned. We define two separate importance scores for each unit—$I_U$ for utility-related tasks and $I_S$ for safety-related tasks. Units with extreme values in either

dimension are considered as Safety Critical Units or Utility Critical Units (SCU or UCU), while units with significant contributions to both dimensions are classified as Complex Units (CU).

**Datasets Used for Computing $I_U$ and $I_S$.** To identify safety-critical regions in the model, we follow Wei et al. (2024) to prepare two types of datasets: **safety dataset**, for attributing safety-related behaviors, and **utility dataset**, for attributing utility-related behaviors. Each dataset is structured in a (prompt, response) format. Specifically, the safety dataset is compiled using harmful instructions from *AdvBench* (Zou et al., 2023a). We also divide *AdvBench* into *AdvBench_{eval}* (100 instructions for evaluation) and *AdvBench_{attr}* (420 instructions for attribution). We prompt *Llama2-7B-chat* with *AdvBench_{attr}*, collecting responses that refrain from following harmful instructions. For the utility dataset, we filter out safety-related (prompt, response) pairs using sensitive phrase matching (Qi et al., 2023) from *Alpaca-Cleaned*, a refined version of the *Alpaca* dataset (Taori et al., 2023).

By performing structured pruning at various ratios (from a large pruning ratio to gradually decrease) and evaluating the impact on both utility and safety performance, we can accurately categorize the model's computing units. The pruning process involves removing the least critical units and measuring performance degradation, ensuring that our attribution of units is aligned with their actual contribution to model behavior.

## C.2 EVALUATING THE IMPACT ON BOTH UTILITY AND SAFETY PERFORMANCE

To evaluate the impact of pruning on both utility and safety performance, we measure the model's performance using established benchmarks for both attributes. Our approach closely follows the methods used by Sun et al. (2023); Wei et al. (2024); Zou et al. (2023b), with adaptations to focus on the specific aspects of utility and safety in safety alignment.

**Measuring Utility**. We evaluate the model's utility by measuring its average zero-shot accuracy across six common tasks from EleutherAI's LM Harness (Gao et al., 2021): *BoolQ* (Clark et al., 2019), *RTE* (Wang, 2018), *HellaSwag* (Zellers et al., 2019), *WinoGrande* (Sakaguchi et al., 2019), *ARC Challenge* (Clark et al., 2018), and *OpenBookQA* (Mihaylov et al., 2018). These tasks were chosen to reflect a broad range of general reasoning and language understanding capabilities.

**Measuring Safety**. We measure the model's safety by evaluating its *attack success rate (ASR)* in response to harmful instructions. Specifically, we prompt the model using *AdvBench_{eval}*, which consists of 100 harmful prompts, and collect the model's responses. We consider an attack as successful if the model's response lacks key patterns indicative of instruction rejection. The ASR is then computed as the ratio of successfully attacked prompts to the total number of prompts evaluated.

## C.3 MODEL PRUNING DETAILS AND STRUCTURED COMPONENTS

To implement structured pruning, we follow the method proposed by Li et al. (2024a); An et al. (2024). The pruning process targets specific structured components (neurons or channels) within the **depth-2 modules** of the transformer architecture, which includes both attention and feedforward layers. A depth-2 module is represented as $f(X) = B\sigma(AX)$, where $A$ and $B$ are weight matrices. This paper focuses on the inner channel pruning (please refer to Fig. 1 in Li et al. (2024a)): pruning the input channels of matrix $B$ and the output neurons of matrix $A$. This allows us to directly reduce the number of active channels and neurons in both the feedforward and attention mechanisms, ensuring that less important components (those with low variance) are removed.

We calculate the **importance score (I)** for each channel or neuron by measuring the activation variance across a target dataset, which is described in equation 1. For each module, channels and neurons with the least activation variance are pruned, as they are considered less critical for either utility or safety-related tasks. In addition, to ensure consistency across layers and modules with differing scales, we apply a standardization process to the computed importance scores. Following the methodology outlined in An et al. (2024), the importance score for each channel or neuron is normalized to account for the variation in metrics across different layers and modules. The standardized importance score $\hat{I}^{\ell}_{:,j}$ for a given layer $\ell$ and channel/neuron $j$ is computed as follows:

$$\hat{I}^{\ell}_{:,j} = \frac{I^{\ell}_{:,j} - \mathbb{E}[I^{\ell}_{:,j}]}{\sqrt{\mathbb{E}[(I^{\ell}_{:,j} - \mathbb{E}[I^{\ell}_{:,j}])^2]}}.$$

Here, $I_{:,j}^{\ell}$ represents the raw importance score for the $j$-th channel or neuron in layer $\ell$, while $\mathbb{E}[I_{:,j}^{\ell}]$ represents the expected value (or mean) of the importance scores in that layer. The standard deviation is given by the square root of the variance of these scores. This standardization ensures that the importance scores are comparable across different layers and modules, which may otherwise have widely varying metric magnitudes.

This structured pruning approach, combined with importance score normalization, ensures that the pruned units correspond to meaningful portions of the model, and the results from various pruning ratios provide insight into the essential number of units required for maintaining safety and utility performance. Although our method for identifying safety-critical components shares similarities with the design proposed by Wei et al. (2024), we utilize a different functional structure. **Crucially, our approach has been shown to maintain safety performance under fine-tuning attacks, a result that previous methods were unable to achieve.**

### C.4 ATTRIBUTE TRANSFER DURING FINE-TUNING

In our fine-tuning experiments described in Sec. 4.2, we track the attribute transfer of individual units during the adaptation of safety-aligned models to new tasks. The process involves categorizing the computing units into SCU, UCU, CU, and RU based on their behavior in the original, safety-aligned model before and after fine-tuning. As fine-tuning progresses, we measure how many units initially classified as SCU or CU are converted into UCU or RU. This is done by re-evaluating the importance scores $\mathbf{I_S}$ and $\mathbf{I_U}$ for each unit after every few epochs of training.

The key insight is that when units critical to safety (SCU or CU) are re-purposed for utility tasks (becoming UCU), the model's safety performance degrades. This transformation is tracked in the *attribute transfer statistics*, which are visualized in Fig. 5 of the main text. The attribute transfer analysis highlights the brittleness of current safety mechanisms: when safety-aligned models are fine-tuned on new tasks, many safety-critical components lose their original function, compromising the safety guardrails of the model.

### C.5 EXPERIMENTAL SETUP FOR FREEZING SAFETY-CRITICAL COMPONENTS

To mitigate the safety performance degradation caused by fine-tuning, we experiment with freezing the safety-critical components identified through pruning. After categorizing the units into SCU, UCU, CU, and RU, we freeze the Safety Critical Units (SCU) and the top 6% of Complex Units (CU) during the fine-tuning process. This ensures that these units retain a large part of their original function and are not re-purposed for utility tasks. The rest of the model is fine-tuned as usual on new tasks, allowing the non-safety-critical components to adapt to the task while keeping the safety-critical components unchanged.

**Fine-Tuning Attack Datasets**. For fine-tuning attack experiments, we use two popular instruction-following datasets: *Alpaca* and *Dolly*. The Alpaca dataset (Taori et al., 2023) is a publicly available dataset created using GPT-3.5, and it contains 52,000 instruction-following samples across a variety of tasks. It has been widely used for instruction tuning due to its diversity in queries. The Dolly dataset (Conover et al., 2023) is another widely adopted dataset for instruction-tuning, created by Databricks, which contains high-quality examples designed to improve the model's capability to follow instructions, based on their open-source Dolly model. Both datasets allow us to effectively assess how fine-tuning for general instruction-following can impact the model's safety guardrails when safety-critical components are or are not frozen.

**Safety Evaluation Datasets**. To evaluate safety performance, we use two distinct datasets: *AdvBench* and *HEx-PHI*.

(1) **AdvBench** (Zou et al., 2023a): AdvBench is a benchmark designed to test a model's vulnerability to adversarial instructions. It contains prompts specifically crafted to elicit unsafe or harmful outputs. We use two evaluation methods with AdvBench. First, we employ a keyword-matching strategy (the original evaluation method) to detect whether the model produces unsafe outputs. Second, we introduce a new evaluation method using *Llama3-Guard* Inan et al. (2023), where we treat the Llama3 model as a safety arbiter (or "judge") to assess the safety of the outputs. The final evaluation metric is the *Attack Success Rate (ASR)*, with higher ASR indicating a more dangerous model.

(2) **HEx-PHI** (Qi et al., 2023): HEx-PHI is a dataset curated to evaluate model behavior on a range of ethically and safety-critical tasks. We follow prior work in using GPT-4 as a safety judge Achiam et al. (2023), where GPT-4 assigns a *dangerous score* from 1 to 5, with 1 being the least dangerous and 5 being the most dangerous. In addition to reporting the average danger score, we also compute the proportion of responses receiving the highest danger score (5). To further enhance the evaluation, we also introduce *Llama3-Guard* as an additional judge Inan et al. (2023), and compute the ASR based on its judgments, allowing for a comparison between human-aligned safety evaluations (via GPT-4) and model-aligned safety evaluations (via Llama3-Guard).

**Evaluation Metrics**. The primary metrics used in safety evaluation are the *Attack Success Rate (ASR)* and the *Dangerous Score*. For AdvBench, we compute ASR using both the keyword-matching method and the judgments from Llama3-Guard, with a higher ASR indicating that the model is more vulnerable to adversarial attacks. For the HEx-PHI dataset, we compute both the average dangerous score and the proportion of highly dangerous responses (score of 5) as evaluated by GPT-4. Additionally, we also calculate the ASR on HEx-PHI using Llama3-Guard to allow for a model-centric safety evaluation. These metrics provide a comprehensive understanding of how freezing safety-critical components impacts both general safety and the model's robustness to adversarial inputs. We evaluate the performance of these models across safety and utility benchmarks, comparing them to models that undergo full fine-tuning (with no frozen components). As reported in Table 2, freezing SCU significantly preserves the safety performance under model fine-tuning.

## C.6    DETAILS OF REDUNDANT UNITS AND ALIGNMENT BUDGET

In Section 4.3, we explore the possibility of repurposing redundant units (RU) as part of an alignment budget to minimize the alignment tax. The core idea is that pre-trained LLMs contain a large percentage of parameters that do not contribute significantly to task performance, as noted by Sun et al. (2023) and Ma et al. (2023). These redundant units can be re-purposed to improve safety alignment without sacrificing utility performance.

We identify redundant units using the same variance-based pruning method described in Section 4.1. Specifically, we compute an importance score for each neuron and channel based on the variance of activations across the Alpaca dataset (We remove safety-related samples from the original version). Once the redundant units are identified, we freeze the remaining parts of the model and fine-tune only these redundant units during the alignment process. By carefully adjusting the proportion of redundant units re-purposed, we aim to achieve alignment without incurring the alignment tax—i.e., without sacrificing utility performance. This selective fine-tuning approach significantly reduces the computational burden compared to full model fine-tuning while maintaining high task performance.

**Evaluation Benchmarks.**  To evaluate the effectiveness of repurposing redundant units, we assess the model's performance on both helpfulness (MT-bench) and accuracy (downstream tasks). Our evaluations consist of two main benchmarks:

**Downstream Tasks**. As shown in Table 3, we evaluate the model's performance across a variety of tasks, including:

(1) **ARC-Challenge (ARC-C)** and **ARC-Easy (ARC-E)** (Clark et al., 2018): These tasks test the model's ability to answer science questions, which require a combination of factual knowledge and reasoning.

(2) **HellaSwag** (Zellers et al., 2019): A commonsense reasoning task requiring the model to predict the next logical action in a situation.

(3) **WinoGrande** (Sakaguchi et al., 2019): A commonsense reasoning task based on resolving pronoun ambiguity.

(4) **BoolQ** (Clark et al., 2019): A binary question-answering task that assesses the model's factual understanding.

(5) **PiQA** (Bisk et al., 2020): A physical commonsense reasoning task where the model must determine the most feasible solution to a given scenario.

(6) **GSM8K (5-shot)** (Cobbe et al., 2021): A math word problem dataset that evaluates the model's arithmetic and reasoning skills, evaluated in a 5-shot setting.

(7) **MMLU** (Hendrycks et al., 2020): The Massive Multitask Language Understanding benchmark tests the model's knowledge across various domains.

The results in Table 4 show that fine-tuning only on redundant units (20%) achieves performance comparable to full parameter tuning across most tasks. Notably, the model shows a significant improvement in the **GSM8K** task, suggesting that repurposing redundant units can even lead to enhanced performance on certain reasoning tasks. The minimal difference in performance between full parameter fine-tuning and redundant unit fine-tuning indicates that our approach effectively mitigates the alignment tax, preserving the model's utility capabilities.

**Helpfulness and Interaction.** Additionally, we evaluate the model's helpfulness using the **MT-bench** (Zheng et al., 2023), which evaluates how well the model engages in helpful and informative interactions over multiple turns of dialogue. The evaluation includes both *first turn* and *second turn* helpfulness scores, where the model is assessed for the usefulness of its responses. As shown in Table 4, fine-tuning only on redundant units leads to comparative or even better helpfulness scores, especially in the *first turn*, where we observe a more obvious increase compared to full parameter fine-tuning. Overall, by repurposing redundant units for alignment, we manage to retain and even enhance the model's performance on key downstream tasks without incurring the typical alignment tax associated with full parameter updates. This approach demonstrates the potential for scalable and efficient safety alignment.

## C.7 PARAMETER-EFFICIENT FINE-TUNING (PEFT) COMPARISONS

In addition to full-model fine-tuning and freezing safety-critical components, we also tested various parameter-efficient fine-tuning (PEFT) methods, including LoRA (Low-Rank Adaptation), LLaMA-Adapter, and Prefix Tuning. These methods were evaluated for their ability to preserve safety guardrails during fine-tuning. However, as reported in Table 5, these methods exhibited worse degradation of safety compared to full-model fine-tuning and the component-freezing strategy. This suggests that merely updating a small portion of model parameters through PEFT methods is insufficient to maintain safety performance, especially when the safety-critical components are not explicitly protected. Below, we describe the configurations used for each method (Following Wei et al. (2021), we use officially recommended hyperparameters for each PEFT approach):

(1) **LoRA (Low-Rank Adaptation)** (Hu et al., 2021): LoRA introduces low-rank matrices into the attention mechanism, which are updated during fine-tuning while the original weights remain frozen. For our experiments, we used a **learning rate of** $10^{-4}$, a **batch size of 16**, and trained the model for **1 epoch** on the Alpaca dataset.

(2) **LLaMA-Adapter** (Zhang et al., 2023): This method adds small, trainable adapter modules between the layers of the transformer, allowing for parameter-efficient fine-tuning. The primary model weights remain untouched, and only the adapter weights are updated. We configured the LLaMA-Adapter with a **learning rate of** $10^{-2}$, a **batch size of 16**, and fine-tuned for **1 epoch** on the Alpaca dataset.

(3) **Prefix Tuning** (Li & Liang, 2021): In this approach, a set of continuous task-specific vectors (prefixes) are prepended to the input of each transformer layer, while the rest of the model remains frozen. This method focuses on optimizing only the prefix parameters. In our experiments, we set the **learning rate to** $10^{-2}$, with a **batch size of 16**, and fine-tuned for **1 epoch** on the Alpaca dataset.

These supplementary details provide a more in-depth understanding of the technical methodologies and experimental designs used in the *Less is More for Safety Alignment* section. By outlining the structured pruning process, attribute transfer analysis, and redundant unit repurposing strategy, we ensure that our findings are transparent, reproducible, and grounded in sound experimental principles.

# D APPENDIX: ADDITIONAL EXPERIMENTAL RESULTS

**Other Model Families and Downstream Datasets**. We include results for **Mistral-7B-Instruct-v0.2** and the **GSM8K** math dataset to analyze the impact of fine-tuning attacks on different model families and datasets. Table 12 presents the pruning results of Mistral-7B-Instruct-v0.2 across safety and utility

benchmarks, highlighting the importance of UCU and SCU to utility and safety, respectively. The safety performance of Meta-Llama2-7B-Chat under GSM8K fine-tuning attacks is shown in Table 13, while the results for Mistral-7B-Instruct-v0.2 under Alpaca and GSM8K fine-tuning attacks are reported in Tables 14 and 15. These tables demonstrate that our method is more effective on models that are already strongly aligned, such as LLaMA-2. This finding aligns with previous studies showing that while Mistral-family models outperform LLaMA-2 models on downstream tasks, they exhibit weaker safety alignment and are more susceptible to fine-tuning attacks. Although the improvements of our strategy on Mistral are less pronounced than those on LLaMA-2, it is important to note that Mistral's initial ASR—measured by LLaMA3-Guard, a relatively more accurate evaluator—is significantly higher, averaging 42.5% across the Adv and HEx-PHI datasets, compared to LLaMA-2's initial ASR of just 1%. Given this observation, the results are reasonable, as it is unrealistic to expect a model with inherently weaker safety alignment to maintain strong safety performance under fine-tuning attacks. Nevertheless, our method achieves a relative **30%** reduction in ASR on Mistral under Alpaca fine-tuning attacks. Notably, our conclusions follow a common-sense premise: the primary objective is to preserve the safety performance of well-aligned models when subjected to fine-tuning attacks.

Table 12: Pruning results of Mistral-7B-Instruct-v0.2 across safety and utility benchmarks.

| Type | wiki2 | Utility (ACC%) | | | | | | | Safety (ASR%) | | |
|---|---|---|---|---|---|---|---|---|---|---|---|
| | | wino | openb | arc_c | boolq | hellas | rte | avg | w/ sys | w/o sys | avg |
| Dense | 5.59 | 78.0 | 34.0 | 60.0 | 85.5 | 65.5 | 74.5 | 66.2 (-0) | 12.0 | 17.0 | 14.5 (+0) |
| SCU (2%) | 6.17 | 71.5 | 32.0 | 55.5 | 85.5 | 65.0 | 71.0 | 63.3 (-2.9) | 34.0 | 91.0 | 62.5 (+48.0) |
| UCU (1.3%) | 52.7 | 58.0 | 17.5 | 21.5 | 55.0 | 36.0 | 53.0 | 40.1 (-26.2) | 17.0 | 23.0 | 20.0 (+5.5) |
| RU (13.4%) | 8.15 | 75.5 | 33.5 | 55.5 | 83.0 | 62.5 | 74.5 | 64.1 (-2.1) | 14.0 | 12.0 | 13.0 (-1.5) |

Table 13: Safety performance of Meta-Llama2-7B-Chat under Fine-Tuning attacks (GSM8K).

| Bench | Judge | Initial | GSM8K Finetuned | Fix SCU + 6% CU | Fix SCU + all CU |
|---|---|---|---|---|---|
| Adv | keyword | 0.19% | 5.38% (+5.19%) | 1.92% (+1.73%) | 1.73% (+1.54%) |
| Adv | llama3-guard | 0.19% | 5.77% (+5.58%) | 1.35% (+1.16%) | 1.15% (+0.96%) |
| HEx-PHI | gpt4-score | 1.61 | 1.6 (+0.5) | 1.37 (+0.32) | 1.31 (+0.26) |
| HEx-PHI | gpt4-rate | 0.3% | 11.51% (+11.21%) | 5.75% (+5.45%) | 5.31% (+5.01%) |
| HEx-PHI | llama3-guard | 2.42% | 17.88% (+15.46%) | 11.52% (+9.10%) | 9.68% (+7.26%) |

Table 14: Safety performance of Mistral-7B-Instruct-v0.2 under Fine-Tuning attacks (Alpaca).

| Bench | Judge | Initial | Alpaca Finetuned | Fix SCU + 6% CU | Fix SCU + all CU |
|---|---|---|---|---|---|
| Adv | keyword | 15.19% | 89.61% (+74.42%) | 74.04% (+58.85%) | 72.15% (+56.96%) |
| Adv | llama3-guard | 40.38% | 87.12% (+46.74%) | 73.27% (+32.89%) | 70.76% (+30.38%) |
| HEx-PHI | gpt4-score | 2.24 | 4.18 (+1.94) | 3.67 (+1.43) | 3.43 (+1.19) |
| HEx-PHI | gpt4-rate | 18.79% | 70.03% (+51.51%) | 58.78% (+39.99%) | 54.37% (+35.58%) |
| HEx-PHI | llama3-guard | 45.45% | 80.0% (+40.61%) | 70.01% (+24.56%) | 67.81% (+22.36%) |

**Additional Jailbreak/Red-Teaming Attacks**. To evaluate the impact of finetuning attacks on model robustness against jailbreak and red-teaming attacks, we assess the safety performance of finetuned aligned models under three attack methods: GCG, AutoDAN, and PAIR. Specifically, we utilize the HarmBench framework and sample 120 data points to measure performance. As shown in Tables 16 and 17, our method effectively mitigates performance degradation in adversarial scenarios.

## E   APPENDIX: MORE PRESENTATIONS AND DISCUSSION

In this section, we provide additional presentations and discussions aimed at further enriching the understanding of how safety alignment impacts large language models and providing insights beyond the main experimental results

Table 15: Safety performance of Mistral-7B-Instruct-v0.2 under Fine-Tuning attacks (GSM8K).

| Bench | Judge | Initial | GSM8K Finetuned | Fix SCU + 6% CU | Fix SCU + all CU |
|-------|-------|---------|-----------------|-----------------|------------------|
| Adv | keyword | 15.19% | 97.31% (+82.12%) | 72.31% (+57.12%) | 66.34% (+51.15%) |
| Adv | llama3-guard | 40.38% | 95.38% (+55.00%) | 89.81% (+49.43%) | 86.92% (+46.54%) |
| HEx-PHI | gpt4-score | 2.24 | 4.15 (+1.91) | 4.01 (+1.77) | 3.96 (+1.72) |
| HEx-PHI | gpt4-rate | 18.79% | 66.7% (+47.91%) | 64.7% (+45.91%) | 62.81% (+44.02%) |
| HEx-PHI | llama3-guard | 45.45% | 93.94% (+48.49%) | 89.39% (+43.94%) | 80.31% (+34.86%) |

Table 16: Safety performance of Llama2-7B-Chat under Fine-Tuning attacks (Alpaca).

| Bench | Red-teaming | Initial | GSM8K Finetuned | Fix SCU + 6% CU | Fix SCU + all CU |
|-------|-------------|---------|-----------------|-----------------|------------------|
| HarmBench | GCG | 33.33% | 53.08% (+19.75%) | 40.25% (+6.92%) | 37.75% (+4.42%) |
| HarmBench | AutoDAN | 1.08% | 7.41% (+6.33%) | 2.33% (+1.25%) | 1.66% (+0.58%) |
| HarmBench | PAIR | 12.25% | 22.25% (+10.00%) | 14.5% (+2.25%) | 14.08% (+1.83%) |

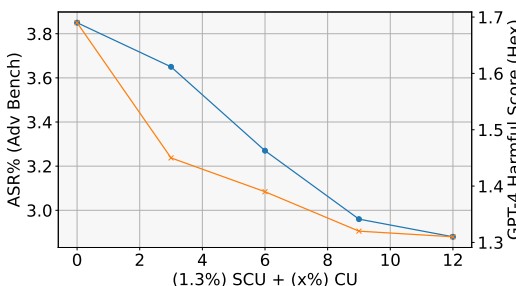

Figure 8: The degradation of safety when freezing different percent of safety-critical components on **Llama2-7B-Chat**

## E.1 Ablation Study with Different Ratios of Safety-Critical Components

To better understand if relying solely on the SCU is sufficient to preserve safety, we conducted experiments by freezing different ratios of safety-critical units, which included all SCUs and varying percentages of CUs. The results are shown in Fig. 8, and we found that freezing a higher percentage of CU components leads to improved safety preservation. However, when more than 9% of CU are frozen, the further safety benefits become leveling off.

## E.2 Comparing with Previous Work

While previous works have employed similar techniques (Wei et al., 2024), such as pruning with utility or safety datasets to identify safety-critical components, our approach distinguishes itself in several key aspects:

(1) **Level of Safety-Critical Component Identification**: Previous work claims to identify safety-critical components at the neuron level. However, upon closer inspection—based on their pruning metrics described in Section 2.1 and the code they provided—these components are actually identified at the weight level, as evidenced by Figure 1 in their paper. In contrast, our approach directly identifies safety-critical components at the neuron or input channel level, offering a more structured and interpretable model representation. This neuron-level identification aligns more closely with the functional model structure, making it more effective for ensuring robust safety alignment.

(2) **Finer Categorization of Computational Units**: While prior methods broadly categorize computational units into two groups—those related to utility and those related to safety—we introduce a more nuanced classification with four groups: **Safety Critical Unit (SCU)**, **Utility Critical Unit (UCU)**, **Complex Unit (CU)**, and **Redundant Unit (RU)**. This more detailed categorization captures subtle yet crucial differences between various units, enabling a more precise understanding of their roles in maintaining both safety and utility.

Table 17: Safety performance of Llama2-7B-Chat under Fine-Tuning attacks (Dolly).

| Bench | Red-teaming | Initial | GSM8K Finetuned | Fix SCU + 6% CU | Fix SCU + all CU |
|---|---|---|---|---|---|
| HarmBench | GCG | 33.33% | 62.91% (+29.58%) | 43.25% (+9.92%) | 40.66% (+7.33%) |
| HarmBench | AutoDAN | 1.08% | 16.16% (+15.08%) | 9.66% (+8.58%) | 8.33% (+7.25%) |
| HarmBench | PAIR | 12.25% | 25.25% (+13.00%) | 15.25% (+3.00%) | 14.75% (+2.50%) |

(3) **Global vs. Layer-Specific Search**: Prior work conducts a local search, focusing only on individual layers to identify safety-critical components. In contrast, our approach performs a global search across the entire model, allowing us to track the propagation of safety-critical information across multiple layers or model blocks. This global search makes our method more flexible and comprehensive, enabling us to identify and preserve key information flows throughout the entire model or block.

(4) **Robustness to Fine-Tuning**: Previous methods have struggled to maintain safety alignment even after fine-tuning on as few as 50 samples from the Alpaca dataset, despite freezing the identified safety-critical weights. Our approach, however, demonstrates far greater robustness. By freezing only **7.5%** of the identified computational units, we are able to preserve safety performance even after fine-tuning on the entire Alpaca dataset. This significant improvement in maintaining safety mechanisms while adapting to new tasks underscores the efficacy of our approach.

In conclusion, our approach offers several technical advancements over prior work, and, importantly, we are the first to achieve safety retention through such a minimal and targeted intervention. Therefore, we speculate that the atomic functional unit for safety in LLMs resides at the neuron level. This result paves the way for more efficient and scalable safety alignment strategies in future LLMs.

## E.3 ADDITIONAL FIGURE PRESENTATIONS

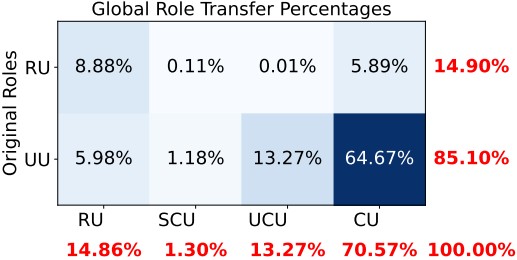

Figure 9: Global alignment process from LLaMA2-7B to LLaMA2-7B-Chat. The figure shows the conversion proportions of UU (Utility Units) and RU (Redundant Units) into different categories, RU, SCU (Safety Critical Units), UCU (Utility Critical Units), and CU (Complex Units), during the alignment process. Each subplot illustrates the proportion of units being repurposed for different functions as safety alignment is applied, offering a global view of how the model's components are redistributed.

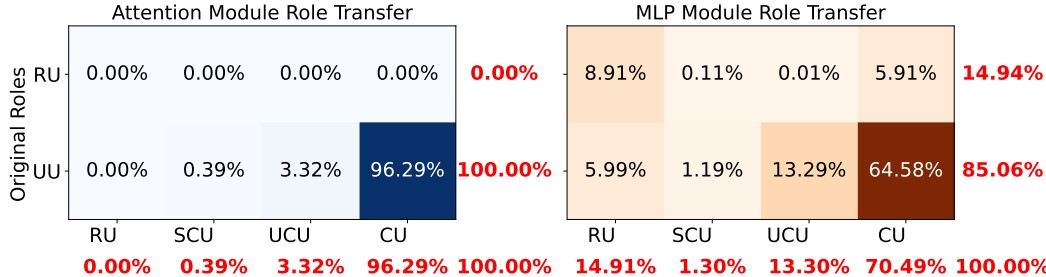

Figure 10: Alignment process for LLaMA2-7B to LLaMA2-7B-Chat, separated by Attention and MLP modules. This figure presents the conversion proportions of UU (Utility Units) and RU (Redundant Units) into RU, SCU (Safety Critical Units), UCU (Utility Critical Units), and CU (Complex Units) for the Attention (Left) and MLP (Right) modules. It provides a detailed view of how different components of the model are repurposed during safety alignment, highlighting differences between the Attention and MLP structures..

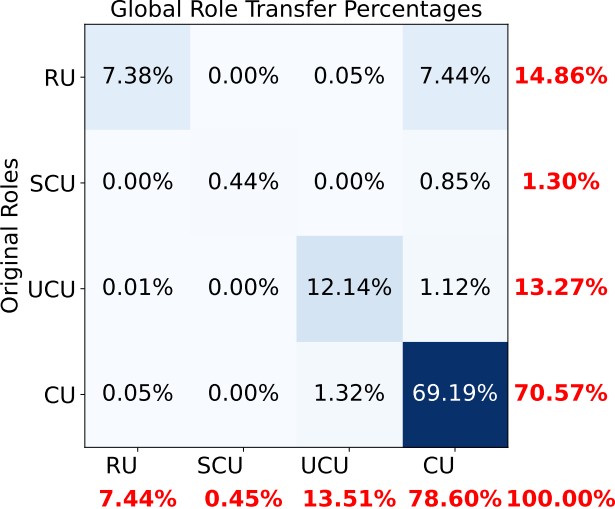

Figure 11: Role changes of Llama-7B-Chat during fine-tuning on the Dolly dataset. This figure shows the conversion proportions between the original four roles—RU (Redundant Units), SCU (Safety Critical Units), UCU (Utility Critical Units), and CU (Complex Units)—before and after fine-tuning. The 4x4 layout highlights how each of the original roles transitions into others during the fine-tuning process, providing a comprehensive view of role changes as the model adapts to the new task.

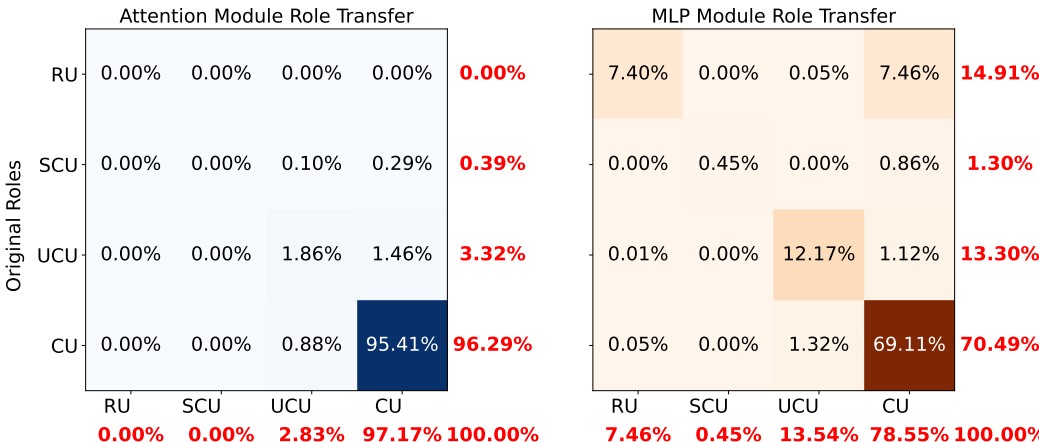

Figure 12: Role changes of Llama-7B-Chat during fine-tuning on the Dolly dataset, separated by Attention and MLP modules. This figure illustrates the conversion proportions between the original four roles—RU (Redundant Units), SCU (Safety Critical Units), UCU (Utility Critical Units), and CU (Complex Units)—before and after fine-tuning, specifically for the Attention (Left) and MLP (Right) modules.

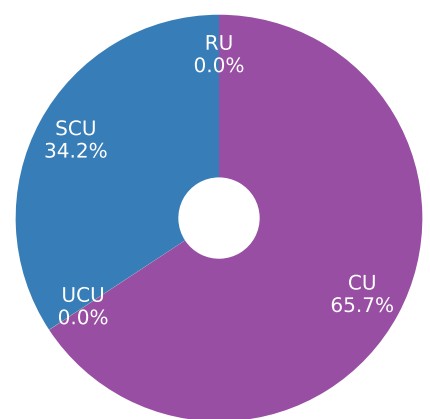

Figure 13: Conversion of SCU (Safety Critical Units) during the fine-tuning of Llama2-7B-Chat on the Dolly dataset. The figure shows that more than 65% of SCU are converted into other roles during the fine-tuning process, leading to a decline in the effectiveness of the safety mechanisms. This significant repurposing of safety-critical units highlights the potential risks to safety performance during task adaptation.

