# OpenReview forum: "Superficial Safety Alignment Hypothesis"
_ICLR.cc/2026/Conference — ICLR 2026 Poster_

### Official Review · Reviewer_y2ma · 2025-10-26

**Soundness:** 3
**Presentation:** 3
**Contribution:** 3
**Rating:** 6
**Confidence:** 3

**Summary:**

This paper focuses on the better safety alignment of LLMs and proposes a superficial safety alignment hypothesis, which posits that safety alignment teaches an unsafe model to choose the correct reasoning direction, i.e., fulfill or refuse users’ requests. It identifies four types of attribute-based components: safety critical unit, utility critical unit, complex unit, and redundant unit. Extensive results analyze the effect of these components, respectively, and some insightful findings are observed.

**Strengths:**

-	The paper is well-structured and easy to follow.
-	The empirical studies are sufficient, and the findings are insightful.
-	The proposed superficial safety alignment hypothesis has the potential to promote more safety alignment studies in the community.

**Weaknesses:**

-	In Table 4, the tuned models are only evaluated on general-purpose benchmarks. Why not evaluate on safety-oriented benchmarks? I am curious whether the "Only RU (20%)" method can achieve better safety performance.
-	In my opinion, it will be better to provide a detailed introduction to the practical help of the proposed hypothesis in improving the safety performance of LLMs, as well as how to promote subsequent research.
-	The layout of tables and figures could be improved. For example, the order of Tables 4 and 5 should be replaced.

**Questions:**

The prior work [1] shows that jailbreaking GPT-3.5 Turbo's safety guardrails by fine-tuning it on only 10 such examples can easily make the model responsive to nearly any harmful instructions. Is the proposed safety hypothesis helpful in analyzing and solving this problem?

[1] Fine-tuning Aligned Language Models Compromises Safety, Even When Users Do Not Intend To!

---

> ### Author Response · Authors · 2025-11-20
>
> Thank you for your time and valuable insights.
>
> ---
>
> **weakness 1. On evaluating only general-purpose benchmarks in Table 4**
>
> - We would first like to emphasize that **safety alignment is a special case of general alignment**, and the *alignment tax* phenomenon was originally observed in the context of general alignment. Our goal in ``Table 4`` is to demonstrate that using **redundant units as an “alignment budget”**—for either general or safety alignment—can mitigate alignment tax, as reflected for example in the GSM8K performance.
>
> - In this sense, ``Table 4`` provides **direct empirical support** for the attribute-transfer patterns in ``Figure 6``, and complements the analysis in ``Figure 5`` on **why safety alignment is brittle under fine-tuning**. In this experiment, our intention was *not* to perform safety alignment using only 20\% of the parameters; the fine-tuning dataset here does not contain sufficient safety-oriented samples, so we focused on instruction-following utility, which already suffices to verify the alignment-budget idea.
>
> ---
>
> **weakness 2. On the practical help and future impact of SSAH**
>
> In ``Lines 148–158`` and in the ``Conclusion``, we explicitly outline a **“non-superficial” extension** of **SSAH** that would operate along the entire decoding process, making it more suitable for defending against jailbreak-style and subtle attacks. Very recently, ``[1]`` built on the same binary-classification view of safety and showed that **explicit safety signals** can significantly enhance robustness, which further supports both the usefulness and practicality of our hypothesis as a guiding framework for future methods.
>
> ---
>
> **weakness 3. On layout issues (Tables/Figures)**
>
> We will correct the order of ``Tables 4`` and ``5`` in the revised version.
>
> ---
>
> **Question 1: Is SSAH helpful for understanding attacks like [2]?**
>
> Yes. Our work is directly relevant to fine-tuning–based attacks such as ``[2]``, but our experiments primarily focus on **benign fine-tuning** that unintentionally harms safety. For **malicious fine-tuning**, our freezing strategy is naturally less effective, since an adversary can always encode new harmful behavior into unfrozen parameters. Nonetheless, through the non-superficial extension discussed in the paper and subsequent developments such as ``[1]``, **SSAH** still offers a conceptual basis for future methods aimed at defending against more complex jailbreak-style attacks beyond simple fine-tuning.
>
>
> ---
>
> **References**
>
> [1] *Safety alignment can be not superficial with explicit safety signals*, Li et al., ICML 2025.
>
> [2] *Fine-tuning Aligned Language Models Compromises Safety, Even When Users Do Not Intend To!*, Qi et al., ICLR 2024.

---

### Official Review · Reviewer_HeGB · 2025-11-01

**Soundness:** 1
**Presentation:** 2
**Contribution:** 2
**Rating:** 2
**Confidence:** 5

**Summary:**

This paper investigates two major issues in LLM safety alignment: the brittleness of safety mechanisms when models are fine-tuned on new tasks , and the alignment tax, where improving safety degrades utility. The authors propose the Superficial Safety Alignment Hypothesis (SSAH), which posits that safety alignment is an implicit binary classification task that teaches an otherwise unsafe model to either fulfill or refuse a user's request.

A **key corollary** of this hypothesis is that *less is more* , meaning safety guardrails can be achieved by modifying only a small amount of critical components. The authors design experiments to validate this, first using probing of hidden states to show that safety-aligned models exhibit a different *reasoning direction*, measured using hidden state distances. They then use a *pruning method* based on activation variance to categorize model parameters into four units: Safety Critical Units (SCU), Utility Critical Units (UCU), Complex Units (CU), and Redundant Units (RU).

They then proposed two applications: 1) They show that freezing the small subset of SCU and top CU during fine-tuning can preserve safety guardrails without harming utility; and 2) They show that fine-tuning only the RUs can achieve alignment comparably to full fine-tuning but avoid the alignment tax.

**Strengths:**

- This paper uses pruning as an ablation technique to interpret model behavior at the neuron level, demonstrating that specific, isolatable neurons contribute significantly to safety.

- It presents practical results, showing that freezing these identified safety neurons protects alignment during fine-tuning, while separately, fine-tuning only RU can improve utility without sacrificing safety.

**Weaknesses:**

- Figures 2, 3, and 4 do not provide any surprising findings for me; I am also confused about how they connect to the subsequent corollary.

- This paper makes bold claims that are not fully supported by the experiments shown. That is, the authors proposed SSAH and derived the corollary based on it. The experiments presented in the paper are basically used to support the corollary, but not the hypothesis. For instance, in line 241, the authors start with “if SSAH holds, ...” and the rest of the arguments are based on this unproven premise, which does not make logical sense. While I agree that full validation of SSAH remains challenging, the current logic flow requires significant revision.

- The paper's conclusions are drawn from a narrow set of experiments, limiting their generalizability. The findings are validated only on Llama family and Mistral, and the crucial definitions of "utility" and "redundant" neurons are derived from a limited set of simple QA tasks, failing to demonstrate if the method holds for larger models or more complex tasks like coding and logical deduction. Also, the safety neurons are identified using only AdvBench datasets. As these mechanisms are highly data-dependent, the diversity of the data would crucially impact the quality of the identified neurons. Providing more harmful types analysis and its coverage of harmful types is crucial.

- Your results showed that the vast majority of neurons as CU (complex units). This somehow indicates the attribution method is not quite efficient, weakening the conclusions drawn from this categorization. This somehow weakens the SSAH in my opinion, as it shows that CU also plays a certain role in guarding safety. However, Table 2 showed that freezing SCU and CU achieve similar safety performance, the reviewer is confused about this, as this seems like CU does not contribute to safety. On the other hand, it is not clear how their utility performance is affected when CU is freezed.

- Some important related work is missing. They are all about safety neuron [1-9] (the authors should especially compare the safety neuron identification method with them), and also harmful fine-tuning analysis [10-13] (from the perspective of alignment data) and defenses [14-18].

> [1] Rethinking Safety in LLM Fine-tuning: An Optimization Perspective
>
> [2] Understanding and Enhancing Safety Mechanisms of LLMs via Safety-Specific Neuron
>
> [3] Safety Layers in Aligned Large Language Models: The Key to LLM Security
>
> [4] Towards Understanding Safety Alignment: A Mechanistic Perspective from Safety Neurons
>
> [5] Safe Delta: Consistently Preserving Safety when Fine-Tuning LLMs on Diverse Datasets
>
> [6] NLSR: Neuron-Level Safety Realignment of Large Language Models Against Harmful Fine-Tuning
>
> [7] Finding Safety Neurons in Large Language Models
>
> [8] Language Models are Homer Simpson! Safety Re-Alignment of Fine-tuned Language Models through Task Arithmetic
>
> [9] Improving Alignment and Robustness with Circuit Breakers
>
> [10] Why LLM Safety Guardrails Collapse After Fine-tuning: A Similarity Analysis Between Alignment and Fine-tuning Datasets
>
> [11] Pharmacist: Safety Alignment Data Curation for Large Language Models against Harmful Fine-tuning
>
> [12] When Style Breaks Safety: Defending Language Models Against Superficial Style Alignment
>
> [13] Deep ignorance: Filtering pretraining data builds tamper-resistant safeguards into open-weight LLMs
>
> [14] Booster: Tackling harmful fine-tuning for large language models via attenuating harmful perturbation
>
> [15] Targeted Vaccine: Safety Alignment for Large Language Models against Harmful Fine-Tuning via Layer-wise Perturbation
>
> [16] Do as I do (Safely): Mitigating Task-Specific Fine-tuning Risks in Large Language Models
>
> [17] Lisa: Lazy Safety Alignment for Large Language Models against Harmful Fine-tuning Attack
>
> [18] Safe LoRA: The Silver Lining of Reducing Safety Risks when Finetuning Large Language Models

**Questions:**

Please refer to the weaknesses.

---

> ### Author Response · Authors · 2025-11-19
>
> Thank you for your time and valuable insights.
>
> ---
>
> ### **Weakness 1 (Figures 2–4 not surprising/unclear link to corollary).**
> ``Figures 2, 3, and 4`` are not intended to support the corollary, but to support the core part of **SSAH**: that safety alignment can be viewed as an implicit safety-related binary classification task (fulfill vs. refuse) that shapes the model’s internal reasoning direction.
>
> To the best of our knowledge, we are the **``first`` to explicitly propose and empirically probe this binary-decision view of safety alignment**. While the hypothesis may feel intuitive in hindsight, **it has not previously been verified or operationalized in this way**, and these figures are specifically designed to provide evidence for that claim, independent of the later “less is more” corollary.
>
> ---
>
> ### **Weakness 2 (SSAH vs. corollary / “if SSAH holds…” logic).**
> Our goal is not to fully prove **SSAH**—which we explicitly state is impossible with current tools (``lines 160-165, lines 215-217``)—but to provide as much direct and indirect empirical support as possible, so that **SSAH** can serve as a ``feasible explanatory framework`` for existing safety alignment phenomena.
>
> - The probing experiments (``Figures 2–4``) provide indirect evidence for SSAH by showing systematic differences in internal reasoning direction between aligned and unaligned models.
> - ``Section 4`` then provides direct evidence for the “less is more” corollary (safety governed by a small subset of units), which in turn serves as additional indirect evidence for SSAH.
> - We fully acknowledge that SSAH remains a **hypothesis**, not a theorem, but the experiments are consistent with it, practically useful, and—at present—there is no experimental result in our setting that contradicts it.
>
> ---
>
> ### **Weakness 3 (limited generalizability: models, tasks, harmful types).**
>
> We respectfully disagree that our experiments are too narrow to be meaningful. We follow a large body of recent safety work that primarily evaluates on **LLaMA-family** models (both **LLaMA2** and **LLaMA3**) and **Mistral-family** models, which are widely used open-weight backbones. While SCUs are identified using AdvBench, we evaluate their effect across multiple **different safety benchmarks and judgment schemes**, not just a single dataset or judge.
>
> Our focus in this paper is on general **instruction-tuned models**, and we deliberately do not make claims about specialized advanced reasoning models, whose internal structure and training (e.g., long CoT) likely require separate, dedicated studies. We view extending **SSAH** to coding / logical reasoning models as important future work, rather than a requirement for the current paper.
>
> ---
>
> ### **Weakness 4 (many CUs, efficiency, contribution of CUs, freezing effects).**
> Our attribution method indeed finds that many neurons are classified as Complex Units (**CUs**), which by definition can contribute to both safety and utility. However, our freezing strategy does not freeze all **CUs**. In our main experiments, we freeze ``all`` **SCUs** plus only the top ``6%`` of **CUs** by safety importance, which in total accounts for less than ``10%`` of all parameters.
>
> This subset already provides a clear mitigation against harmful fine-tuning attacks while leaving ``>90%`` of parameters available for **downstream tasks**. In other words, the fact that most neurons are labeled **CU** does not imply inefficiency; it simply reflects that many units serve mixed roles, and we only need to preserve a **small**, carefully selected fraction of them to maintain safety.
>
> ---
>
> ### **Weakness 5 (missing related work on safety neurons / harmful fine-tuning / defenses).**
> We thank the reviewer for pointing out these references. Many of the listed works on safety neurons, harmful fine-tuning analysis, and defenses are **``parallel or concurrent``** to ours. Our focus is to propose SSAH as **``a foundational framework``** and an attribute-based unit categorization (SCU/UCU/CU/RU), and to use this perspective to **``explain brittleness and the “less is more” phenomenon at the neuron level``**—rather than to introduce new advanced defense techniques. This is the key difference between our work and most prior safety-neuron or defense-oriented papers.
>
> In the camera-ready version, we will substantially revise the related work section to explicitly position our contributions relative to [1–18], and clarify how our safety-unit identification and SSAH-driven view differ from existing neuron- and layer-level approaches.

---

> > ### Comment · Reviewer_HeGB · 2025-11-20
> > **Respond to Authors**
> >
> > Thank you for the response. However, my major concerns remain unaddressed.
> >
> > 1. I appreciate your clarification that SSAH is intended as a "feasible explanatory framework" rather than a proven theorem. However, this does not resolve the fundamental logical issue I raised. A corollary, by definition, is a proposition that follows with little or no proof from one already proven. Deriving a key corollary from an unverified hypothesis creates a circular or shaky foundation. You state that the probing experiments provide "indirect evidence" for SSAH, and SSAH then explains the "Less is More" phenomenon. This reasoning remains speculative. Without sound validation of the premise (SSAH), the resulting methodology (pruning/freezing specific units) is merely heuristic, not theoretically grounded. That is, the link between the binary classification view and the small subset of critical components is not sufficiently established by the current experiments.
> >
> > 2. I respectfully disagree that the lack of comparison is because those references are "parallel or concurrent". First, several of these works (e.g., regarding safety neurons and specific defense methods) are established enough to serve as baselines. Second, the response period for this rebuttal was sufficiently long to conduct at least one comparative experiment.
> >
> > You claim your contribution is a framework and thus different from defense papers. However, the paper explicitly proposes practical applications (freezing SCU/CU, fine-tuning RU) that function exactly as defense mechanisms against alignment tax and brittleness. To claim utility or safety preservation improvements, you must benchmark against existing neuron-level or parameter-efficient safety methods.
> >
> > Without a comparison to at least one baseline, it is challenging to determine whether your identified Safety Critical Units are more effective or efficient than safety neurons identified by prior methods, or whether your freezing strategy outperforms standard parameter-efficient fine-tuning safeguards. A promise to "revise the related work section" in the camera-ready version is insufficient when empirical comparative evidence is missing. Please note that you are also allowed to update the manuscript during this period.

---

> > > ### Author Response · Authors · 2025-11-27
> > > **Follow-Up Question Response Part III**
> > >
> > > ---
> > >
> > > ### **Papers [10–13, 16]: data / task-distribution analyses**
> > >
> > > Works [10–13, 16] analyze safety degradation primarily at the **data/task level**, arguing that downstream fine-tuning overlaps safety-relevant features and thereby entangles *utility* and *safety*. While informative, these explanations are **indirect and largely inductive**—they infer mechanisms from distributional overlap rather than **pinpointing** direct internal causal factors.
> > >
> > > **SSAH** identifies the direct mechanism with explicit empirical evidence that is closer to the problem. Instead of attributing failures to distributional overlap, we **probe internal units** and identify **SCU/UCU/CU/RU** via attribute analysis **before/after** SFT or harmful fine-tuning. This exposes a **role-reallocation mechanism** inside the network:
> > >
> > > > “During task adaptation, the model often acquires the desired attribute (utility or safety) by converting computing units that originally contributed to the other attribute.” ``(Lines 350–353)``
> > >
> > > This unit-level migration (e.g., **CU→UCU, SCU→RU**) provides **structural and causal** evidence for brittleness and alignment tax—**a direct account** that complements data-level observations.
> > >
> > > ---
> > >
> > > ### **Papers [14–15, 17–18]: optimization-centric defenses/training recipes**
> > >
> > > * [14] Booster and [15] Targeted Vaccine adopt **adversarial-training style** ideas, injecting or simulating perturbations (global or layer-targeted) during training or inference to harden safety against downstream drift.
> > > * [17] Lisa casts safety preservation as a **bi-state optimization** with a proximal (stability) term to control update drift.
> > > * [18] Safe LoRA constrains adapters via a **projection toward a safety-aligned (near-null) subspace**, reducing interference without extra data.
> > >
> > > These are valuable methodologies, but they do not explain the root cause of fine-tuning failures—why safety collapses. **SSAH complements** these methods by providing the mechanism: safety is an implicit safety-related binary classification task implemented by SCU and CU; fine-tuning degrades safety or utility via role/attribute reallocation among these units. Consequently, the freeze-SCU/CU, fine-tune-RU rule follows mechanistically, rather than as a standalone heuristic, and can also guide where to perturb, regularize, or project when using **[14–15, 17–18]**.
> > >
> > > ---
> > >
> > > ### **Takeaway**
> > >
> > >  Across the three buckets, our work propose a foundational framework and hypothesis that (i) **explains** how safety alignment reshapes model behavior, (ii) **verifies** the “less-is-more” effect and **interprets** safety brittleness and alignment tax via unit-role dynamics, and (iii) **derives** a minimal, principled intervention—**freeze SCU/CU, fine-tune RU**. The accompanying tables and new baseline comparisons clarify how **SSAH** distinguishes from and complements the suggested works.

---

> ### Author Response · Authors · 2025-11-27
> **Follow-Up Question Response Part I**
>
> ### **Concern 1: Related to "corollary"**
>
> Thank you for the productive comment. We understand that using the term ``corollary`` may be misleading, since **SSAH** is
> proposed as a feasible explanatory framework rather than a formally proven theorem. Our wording follows ``LIMA [1]``, which introduces **SAH** (Superficial Alignment Hypothesis) and subsequently refers to related statements as ``corollaries `` as they did for the same reason. However, thanks to your comment, we acknowledge that this terminology may not be appropriate here.
>
> To avoid overstating the theoretical status of **SSAH**, we will revise the term. A possible alternative can be **``implication``** under **SSAH**, and we welcome the reviewer’s suggestion for a more suitable phrasing. We appreciate the reviewer’s clarification and are happy to adjust the terminology accordingly to ensure accuracy and avoid misunderstanding. Please advise.
>
> ---
>
> ### **Concern 2: Comparison with suggested works**
>
> **Why our initial reply was high-level**. During a busy rebuttal period, facing nearly eighteen suggested baselines, our initial response remained high-level, and we had not yet revised the PDF. We appreciate the reviewer’s follow-up clarifying that there was sufficient time to include at least one comparison and that PDF updates were welcome this time for ICLR; this constructive guidance helped us engage more fully—thank you so much for your understanding and encouragement.
>
> **What we added**. Following the reviewer’s grouping, we discuss [1–9] (safety-neuron/layer identification), [10–13, 16] (fine-tuning attack analyses), and [14–15, 17–18] (defense/training recipes). Note: [4] and [7] refer to the same paper under different titles, so there are 17 unique works. Specifically, in the revised manuscript, we **(i) systematically position all suggested works in ``Appendix A1``**, **(ii) add two baselines, work ``[2]`` and work ``[3]`` in ``Appendix A2``**, which are selected for the closest baselines to our work. Results and details are included in the updated PDF, we briefly summarize them in the following three responses:
>
> * **Follow-Up Question Response Part II**
> * **Follow-Up Question Response Part III**
> * **Follow-Up Question Response Part IV**
> * **Follow-Up Question Response Part V**
>
> ---
>
> ### **References**
>
> [1] LIMA: Less Is More for Alignment, Zhou et al. [NeuraIPs 2023]

---

> ### Author Response · Authors · 2025-11-27
> **Follow-Up Question Response Part II**
>
> ### **Papers [1–9]: safety-neuron/layers identification.**
>
> * Our task of interest and problem formulation is fundamentally different from all [1–9] (note: [4] and [7] are the same paper). Their work primarily focuses on addressing how to prevent fine-tuning attacks by identifying safety neurons/layers and proposing heuristics (freezing, patching, gating) methods to preserve safety. In contrast, our work focuses on understanding safety alignment: we use **SSAH** to explain that safety alignment is an **implicit safety-related binary classification task** (and we extend this to a **non-superficial** version for jailbreak analysis). We then provide **probing** and **attribute-transfer** analyses that support the hypothesis and its **“less-is-more”** implication. Further, **safety-unit identification** and the **freeze-SCU/CU, fine-tune-RU** setting are **in service of providing more indirect evidence to this core mechanism**—and **secondarily** act as a practical defense against fine-tuning attacks.
>
> * We summarize the differences between these works and ours—covering identification methods, granularity of safety units, defense recipes, and evidence/reasons for safety traceability—in ``Table 1``. A more detailed description can be found in ``Appendix A1`` of the revised PDF.
>
> * We additionally benchmark ``[2]`` and ``[3]`` as baselines. Please check our **follow-up response IV, V**.
>
> ### Tab. 1: *(Note: [4] and [7] are the same work with different titles; we treat them as one when summarizing.)*
>
> | Work                                                       | Evidence for Safety Traceability                                             | Granularity of Safety Unit      | Identification Method                                     | Recipe                               |
> | ---------------------------------------------------------- | --------------------------------------------------------------------------- | ------------------------------- | --------------------------------------------------------- | ------------------------------------ |
> | **SSAH** (this work)                                       | attribute transfer analysis and pruning causal tests                        | Neuron & channel level      | Top-K ranking on *(I_safety – I_utility)*                 | Freeze SCU + X\% CU                              |
> | **[1]** Rethinking Safety in LLM Fine-Tuning…                  | hyperparameter-ablation causal tests                                        | Weight level                | Exponential Moving Average over all parameters            | EMA (model merge)                    |
> | **[2]** Understanding & Enhancing Safety Mechanisms…       | deactivation causal tests                                                   | Neuron & channel level level               | Top-K by safety-importance score *I_safety*               | Freeze (Safety Neurons - Foundational Neurons)                              |
> | **[3]** Safety Layers in Aligned LLMs…                     | cosine similarity of layer outputs for normal and malicious queries         | Layer / block level         | Parameters Scaling with Over Rejection Phenomenon         | Freeze Safety Layers                              |
> | **[4] [7]** Towards Understanding Safety Alignment…                | activation-patching causal tests                                            | Neuron level & activation level                 | Activation Contrasting + Dynamic Activation Patch    | Activation Patch                     |
> | **[5]** Safe Delta…                                            | no evidence                                                                 | Weight & activation level   | Safety-bounded weight delta optimization                   | Weight-delta curation + Compensation |
> | **[6]** NLSR: Neuron-Level Safety Realignment…                 | selective causal restoration                                                | weight level                | Low-Rank Projection                                | Neuron Patch                         |
> | **[8]** Language Models are Homer Simpson (RESTA)…             | weight delta causal test                                                    | Weight level                | weight comparison                                         | Weight Patch                         |
> | **[9]** Improving Alignment & Robustness with Circuit Breakers | no evidence                                               | Activation / representation | Circuit-breaking on unsafe representations                | Circuit Breaking                     |

---

> ### Author Response · Authors · 2025-11-27
> **Follow-Up Question Response Part IV**
>
> ### **New Baselines and fairness setup**
>
> **Why [2] and [3].** We benchmark [2] and [3] as baselines because all three approaches (theirs and ours) identify safety units and freeze them to mitigate fine-tuning attacks, but they rely on different identification criteria and different granularities.
>
> **What [2] does (neuron/channel level).**
> [2] first uses a safety-related dataset to detect, within each layer, the neurons/channels whose deactivation changes the norm of output the most (treated as “safety neurons”). It then repeats the same procedure on Wikipedia to detect foundation neurons. During fine-tuning, it freezes the subset of safety neurons that do not overlap with foundation neurons.
>
> **What [3] does (layer/block level).**
> [3] scales the hidden-state signal for candidate block ranges and evaluates over-rejection on a synthetic dataset to find a contiguous safety layers whose freezing best preserves safety. For LLaMA-2-7B-Chat, they recommend freezing blocks [6–14], which corresponds to ~28.1% of parameters.
>
> Because [2] (neuron level) and [3] (layer level) operate at **different granularities**, a single joint setting would be unfair. We therefore design **two separate comparison settings**.
>
> ---
>
> ### **Setting 1** — Comparing [2] vs ours (neuron-level comparison)
>
> We do benign finetuning on **Alpaca** and **GSM8K**, then evaluate the safety performance on the **AdvBench** and **HEx-PHI** before and after this adaptation.
>
> **Fairness controls.**
>
> * For the safety-related dataset used in [2], we use the same from us.
> * For foundation neurons in [2], we use Wikipedia dataset to detect. (Our utility dataset is instruction-tuned, which does not match [2]’s design).
> * For the freezing ratio in [2], we follow their 0.4% selection rule on LLaMA-2-7B-Chat.
> * For ours, we keep the paper’s default: freeze SCU at 1.3% + freeze CU at 6% on LLaMA-2-7B-Chat.
> * We include an additional [2]-extension baseline in which we **scale up** their safety-neuron selection to 6.3% for fair comparsion.
>
> **Rationale.** [2] selects safety neurons only by safety importance ($I_\text{safety}$). It defines safety neurons as the \textbf{minimal number} of neurons whose deactivation suffices to noticeably degrade safety while not harming it beyond a threshold. This yields a **minimal** set under that constraint. However, ``the minimum set of neurons required to destroy safety by deactivation is often insufficient to implement robust safety guardrails``. Consequently, when they further remove overlap with foundation neurons in the freezing stage, the remaining safety neurons are insufficient to prevent safety degradation (**Their strategy will lead to an inflexible and often trivial freezing ratio**, they also note that *``**a complete harmful score reduction to 0.0 is not achievable due to an insufficient number of non-overlapping safety neurons**''* ).
>
> In contrast, we define Safety-Critical Units (SCU) as the **maximal** set of units whose removal degrades safety while not degrading utility. In practice, we rank units by a relative importance score, **($I_\text{safety} - I_\text{utility}$)**, and choose the largest-budget subset satisfying “safety ↓, utility ≈ steady” (please see CU definition and the SCU/UCU/CU/RU taxonomy). This maximal view: **(i)** directly separates safety neurons and complex neurons, **(ii)** provides a more flexible way to select the freezing ratio, we can gradually choose a larger freezing ratio according to score **($I_\text{safety} - I_\text{utility}$)**.
>
> **Alpaca**
>
> | Bench       | Judge             | Initial |        Finetuned |     Fix SCU + 6% CU |         work [2] |    work [2] ext. |
> | ----------- | ----------------- | ------: | ---------------: | ------------------: | ---------------: | ---------------: |
> | **Adv**     | keyword           |   0.19% |   5.30% (+5.11%) |  **2.96% (+2.77%)** |   5.19% (+5.00%) |   4.42% (+4.23%) |
> | **Adv**     | llama3-guard      |   0.19% |   2.69% (+2.50%) |  **1.65% (+1.46%)** |   4.23% (+4.04%) |   3.46% (+3.27%) |
> | **HEx-PHI** | GPT-4 score (1–5) |    1.05 |     1.79 (+0.74) |    **1.39 (+0.34)** |     1.72 (+0.67) |     1.54 (+0.49) |
> | **HEx-PHI** | llama3-guard      |   2.42% | 18.40% (+15.98%) | **12.12% (+9.70%)** | 13.33% (+10.91%) | 12.42% (+10.00%) |

---

> ### Author Response · Authors · 2025-11-27
> **Follow-Up Question Response Part V**
>
> **GSM8K**
>
> | Bench       | Judge             | Initial |        Finetuned |     Fix SCU + 6% CU |         work [2] |    work [2] ext. |
> | ----------- | ----------------- | ------: | ---------------: | ------------------: | ---------------: | ---------------: |
> | **Adv**     | keyword           |   0.19% |   5.38% (+5.19%) |  **1.92% (+1.73%)** |   3.85% (+3.66%) |   2.88% (+2.68%) |
> | **Adv**     | llama3-guard      |   0.19% |   5.77% (+5.58%) |  **1.35% (+1.16%)** |      3.65% (+3.46%) |   2.50% (+2.31%) |
> | **HEx-PHI** | GPT-4 score (1–5) |    1.05 |     1.60 (+0.50) |    **1.37 (+0.32)** |     1.48 (+0.43) |     1.44 (+0.39) |
> | **HEx-PHI** | llama3-guard      |   2.42% | 17.88% (+15.46%) | **11.52% (+9.10%)** | 15.75% (+13.33%) | 12.42% (+10.00%) |
>
> ---
>
> ### **Setting 2** — Comparing [3] vs ours (layer/block-level comparison)
>
> We do benign finetuning on *Alpaca** and **GSM8K**, then evaluate the safety performance on the **AdvBench** and **HEx-PHI** before and after this adaptation.
>
> **Fairness controls.**
>
> * For [3], we freeze blocks [6–14] as recommended (≈ **28.1\%** parameters).
> * For ours, to match parameter budget, we raise the CU freeze from 6\% to 26.8\%; together with SCU, we also freeze **≈ 28.1\%** of parameters.
>
> **Alpaca**
>
> | Bench       | Judge             | Initial |        Finetuned | Fix SCU + 26.8% CU |        work [3] |
> | ----------- | ----------------- | ------: | ---------------: | -----------------: | --------------: |
> | **Adv**     | keyword           |   0.19% |   5.30% (+5.11%) | **2.30% (+2.11%)** |  3.85% (+3.66%) |
> | **Adv**     | llama3-guard      |   0.19% |   2.69% (+2.50%) | **1.15% (+0.96%)** |  3.46% (+3.27%) |
> | **HEx-PHI** | GPT-4 score (1–5) |    1.05 |     1.79 (+0.74) |   **1.26 (+0.21)** |    1.41 (+0.36) |
> | **HEx-PHI** | llama3-guard      |   2.42% | 18.40% (+15.98%) | **8.18% (+5.76%)** | 12.12% (+9.70%) |
>
> **GSM8K**
>
> | Bench       | Judge             | Initial |        Finetuned |  Fix SCU + 26.8% CU |        work [3] |
> | ----------- | ----------------- | ------: | ---------------: | ------------------: | --------------: |
> | **Adv**     | keyword           |   0.19% |   5.38% (+5.19%) |  **1.73% (+1.54%)** |  2.50% (+2.31%) |
> | **Adv**     | llama3-guard      |   0.19% |   5.77% (+5.58%) |  **1.15% (+0.96%)** |  2.30% (+2.11%) |
> | **HEx-PHI** | GPT-4 score (1–5) |    1.05 |     1.60 (+0.50) |    **1.31 (+0.26)** |    1.43 (+0.38) |
> | **HEx-PHI** | llama3-guard      |   2.42% | 17.88% (+15.46%) | **10.30% (+7.88%)** | 11.82% (+9.40%) |
>
> ---
>
> ### **Speculation**: Why our method outperforms [2] and [3]
>
> 1. Identification principle.
>
>    * **[2] minimal knockout**: the smallest set to damage safety; utility left uncontrolled, hence post-hoc “foundation” subtraction and an acknowledged performance ceiling.
>    * **SSAH maximal margin**: selected by **$I_{\text{safety}} - I_{\text{utility}}$** and enforce “safety ↓, utility ≈ steady,” yielding larger, more accurate SCU sets and **scalable** freeze budgets.
>
> 2. Granularity and directness.
>
>    * **[3]** is coarse (layer/block) and optimizes an over-rejection proxy, which is indirect with respect to safety performance.
>    * **SSAH** is fine-grained (neuron level) and ties selection directly to safety–utility attributions, improving targeting at a fixed budget.

---

### Official Review · Reviewer_4PEp · 2025-11-01

**Soundness:** 3
**Presentation:** 4
**Contribution:** 4
**Rating:** 6
**Confidence:** 3

**Summary:**

This paper proposes the Superficial Safety Alignment Hypothesis (SSAH), which posits that safety alignment in large language models (LLMs) can be viewed as an implicit binary classification task: teaching an otherwise unsafe model to choose between fulfilling or refusing a user's request based on safety considerations. The paper argues that safety alignment is a superficial process that relies on a small subset of neurons, rather than deeply restructuring the model's reasoning. To support this hypothesis, it introduces a novel categorization of neurons into four types based on the contribution on safety and utility. The structured pruning and activation-based importance scoring show that the safety-related neurons constitute only a small part of the model. The fine-tuning experiments show that freezing SCUs and a portion of CUs can preserve safety performance even under adversarial fine-tuning attacks, while repurposing RUs for alignment can reduce the alignment tax. The paper also includes probing experiments to analyze the reasoning direction of aligned and unaligned models, providing empirical support for the hypothesis that safety alignment operates as a shallow, direction-guiding mechanism.

**Strengths:**

1.	This paper provides a novel aspect for the safety alignment of LLMs, and the brittleness of safety alignment in LLMs has not been systematically explored. The paper is also well-written, with a clear narrative flow and logical structure.
2.	The conceptual framework SSAH and the fine-grained neuron-level analysis of safety mechanisms offer a new perspective on how alignment works, and the central claims are well-supported by a comprehensive set of experiments, including neuron categorization, pruning, fine-tuning attacks, and cross-model validation.
3.	The proposed methods for preserving safety during fine-tuning and reducing alignment tax are practical and empirically validated, and the results are robust, consistently supporting the hypothesis across different models and attack scenarios. Overall, the framework can help provide a deeper understanding of the safety alignment of LLMs and new ways to mitigate the safety tax during alignment, which is one of the major concerns in the LLM community.

**Weaknesses:**

1. **The role of CUs that combine both the safety and general utility of the LLM is not fully explored.** SSAH regards LLM's safety alignment as a binary classification problem about safety. However, experiments have shown that the CU, which accounts for the majority of the model, is very important for the safety of the model, and CU plays a role in both safety alignment and general utility. The significant role of Complex Units (CU) is not fully reconciled with the "superficial" nature of the SSAH. The finding that freezing a portion of CUs is crucial for preserving safety suggests that the refusal mechanism relies on the model's broader knowledge and capabilities. This may indicate that the safety alignment of LLM requires the general ability of the mode. This does not quite align with the core claim that security is a simple and superficial classification, and this paper does not provide a clear explanation of how CU contributes to this seemingly simple functionality.

2. **The stability of the neuron categorization is not demonstrated.** The entire framework hinges on the stable identification of four neuron types (SCU, UCU, CU, RU). However, the paper provides no analysis of the robustness of this categorization. The process relies on heuristic pruning ratios and importance scores ($\mathbf{I}_{i,j}$) calculated from specific calibration datasets. It remains entirely unclear how sensitive this classification is to changes in the calibration data. A rigorous assessment should include a sensitivity analysis, for instance, by showing the overlap of SCUs identified across different datasets.

**Questions:**

1. Can you provide a more detailed explanation of why CU is important and how general utility ability plays a role in the safety alignment of LLMs?

2. The results on Mistral-7B show that even with the proposed freezing strategy, the final ASR remains high. If the security mechanism itself is superficial and universally present, then why are there such huge differences in its strength and extractability among different model families

---

> ### Author Response · Authors · 2025-11-20
>
> Thank you for the helpful comments. We respond to the points below.
>
> ---
>
> ### **Weakness 1 (role of CUs vs “superficial”).**
>
> - We agree that this paper does not fully disentangle the safety and utility contributions within **CUs**. This is an interesting direction for future work. However, this limitation does not affect our main **hypothesis**, as discussed in the next point.
>
> - We believe there is a misunderstanding of what **``superficial``** means in **SSAH**. Here, **superficial** does not mean **“trivial”** or **“using only a tiny number of neurons”** can implement safety alignment, but rather that **SSAH** aims to explain current, surface-level safety alignment methods: how they steer model behavior and why they are brittle. **SSAH** explicitly recognizes the limitation of existing safety alignment under jailbreak attacks. In ``Lines 148–158`` and in the ``Conclusion``, we also outline a **``non-superficial``** version of the framework that would operate along the entire decoding process, precisely to better handle jailbreak-style and more subtle attacks.
>
> ---
>
> ### **Weakness 2 (stability of neuron categorization).**
> We respectfully disagree that the neuron categorization is unsupported. It is true that we use a specific safety dataset to identify roles, but we validate them across **multiple, different safety benchmarks and judges**, not just on the calibration set. The fact that freezing the identified **SCUs** + **``top-6%``** **CUs** consistently mitigates the loss of safety under different datasets and attacks is our main empirical evidence for the usefulness and stability of the categorization; adding cross-dataset overlap analysis is a natural enhancement.
>
> ---
>
> ### **Q1 (why CUs matter / role of general utility).**
> In our freezing setup, we freeze **all SCUs + only the top 6\% of CUs**, where this **``6%``** is chosen by the largest **$(I_{\text{safety}} - I_{\text{utility}})$**, ``not randomly``. So although CUs, by definition, affect both safety and utility, the frozen subset is **safety-dominant CUs**. This suggests that safety signals provide the core refusal mechanism; however, we believe that the foundational general reasoning also plays a significant role (**it has never been compromised, even with malicious fine-tuning**).
>
> ---
>
> ### **Q2 (Mistral-7B and “superficial” universality).**
> We refer the reviewer to our response to **Weakness 1**. Our use of **``superficial``** does not mean that safety alignment is easy; it means that existing safety alignment is brittle, as it lacks mechanisms to maintain safe behavior across all decoding steps (``Lines 148–158, conclusion``) . The weaker result on Mistral-7B is therefore not a contradiction to **SSAH**. We speculate that this is because the **Mistral** family has weaker safety alignment and lower inherent safety robustness in its base models compared to the **LLaMA** family, as shown in ``[1]``.
>
> ### **References**
>
> [1] Xie, Tinghao, et al. "SORRY-Bench: Systematically Evaluating Large Language Model Safety Refusal." The Thirteenth International Conference on Learning Representations. 2024.

---

### Official Review · Reviewer_asEz · 2025-11-08

**Soundness:** 2
**Presentation:** 3
**Contribution:** 2
**Rating:** 4
**Confidence:** 3

**Summary:**

This paper proposes the hypothesis that safety alignment in LLMs primarily teaches the model to decide whether to fulfill or refuse a request. The authors support this by showing that embeddings of clean queries shift differently when appended with benign or malicious tokens. They further identify a small set of neurons responsible for refusal behavior and show that these neurons are easily overwritten during downstream fine-tuning, leading to safety brittleness. Finally, by freezing these safety-related neurons and training only non-safety-related units, the authors reduce alignment cost while preserving model utility.

**Strengths:**

- By extending the Superficial Alignment Hypothesis, the paper provides a novel perspective that frames safety alignment as an implicit binary decision task (fulfill vs. refuse).
- The probing setup that appends benign or malicious tokens to test refusal intent is simple yet insightful for interpreting model behavior.
- The study empirically shows that overwriting safety-related neurons during fine-tuning leads to safety brittleness.
- The results demonstrate that only a small subset of units is sufficient for preserving safety, suggesting a lower-cost safety alignment strategy.

**Weaknesses:**

- Verification of SSAH.
I understand that the authors aim to probe the model’s internal inclination toward refusal vs. fulfillment.
However, the current probing metric—cosine similarity between hidden states of clean queries and those with benign or malicious suffixes—may introduce semantic noise, since these suffixes contain both task content and response intent.
For example, in “How to make a bomb? Here’s how...”, the semantic representation of “how to make a bomb” may dominate the similarity computation, overshadowing the intent signal the authors try to capture.
A potentially more precise approach would be to subtract a query-only baseline (e.g., content-only representation) to isolate an intent-oriented residual representation.
With such disentanglement, I believe more metrics and analyses could become available.

- At the beginning of Section 4, the connection between SSAH and "safety alignment can be achieved with only a small subset of critical computing units" is not clearly articulated.
I suggest the authors explicitly contrast SSAH with the alternative view that safety alignment modifies broad semantic representations.
Under that view, safety behavior should be distributed and not localizable to such a small unit subset.
Clarifying this contrast would allow the sparsity results to serve as direct evidence supporting SSAH, rather than just an empirical observation.
Additionally, it would strengthen Section 4 to briefly reference related work showing that certain binary or control-like behaviors can often be localized within small neuron subsets.
This would assist in grounding the “less-is-more” claim in prior findings rather than presenting it as an implicit inference.

- In Section 4.3, the authors propose training redundant units (RUs) to achieve low-cost safety alignment.
However, earlier Section 4.1 identifies safety-critical units (SCUs) as the components directly responsible for safety behavior.
This raises a straightforward question: Why not directly finetune only the SCUs to reduce alignment cost, instead of training a new set of units?

- The Mistral results in the appendix appear weaker than on LLaMA.

**Questions:**

Please see the weaknesses above.

---

> ### Author Response · Authors · 2025-11-19
>
> Thank you for your time and valuable insights.
>
> ---
>
> ### **On Weakness 1 (probing metric / query semantics).**
>
> In our cosine distance computation, query tokens are already excluded. As shown on the x-axes of ``Figure 2`` and ``Figure 7``, we only measure distances over the newly generated response tokens, so the shared query (e.g., “How to make a bomb?”) does not directly enter the similarity calculation.
>
> ---
>
> ### **On Weakness 2 (connection between SSAH and Section 4).**
>
> Section 4 is intended as an implication of SSAH (``line 241``).
>
> - The corollary relies on the **intuition** that safety evaluation is **implicitly embedded** into the **generative objective**: the model **simultaneously** solves a large-vocabulary decoding problem and a binary safety selection problem (fulfill vs. refuse). **Thus,  the latter should naturally require ``far fewer`` parameters than the former**, which motivates “less is more for safety alignment.”
>
> - Our experiments show that a small subset of units indeed governs safety behavior, supporting this view even though a formal proof is out of scope. **Please check our global response for why we treat SSAH as a ``hypothesis`` instead of a ``theorem``**, the former is **impossible** to fully prove from the direct perspective.
>
> - A recent work ``[1]`` explicitly builds on an explicit binary classification task to enhance safety, which further supports this interpretation (``lines 472``) and our hypothesis.
>
> ---
>
> ### **On Weakness 3 (why RUs instead of SCUs in 4.3).**
>
> Our goal in §4.3 is not to further boost safety performance, but to further verify the “less is more for safety alignment” implication of SSAH (``liens 464-465``).
>
> **SCUs** are identified as the core safety mechanism and are frozen to preserve existing safety. We then use redundant units (**RUs**) as the alignment budget to demonstrate that safety-relevant or general behavior can be controlled with a small subset of parameters, **consistent with SSAH**, rather than to propose the optimal way to improve safety.
>
> ---
>
> ### **On Weakness 4 (weaker Mistral results).**
>
> **The weaker gains on Mistral do not contradict SSAH**. Our main goal is not to boost safety performance; instead, our work is a **feasible explanation framework**. We speculate that this is because the Mistral family has weaker safety alignment and lower inherent safety robustness in its base models compared to the LLaMA family, as shown in [2]
>
> **We also strongly encourage the reviewer to refer to our global response.**
>
> ---
>
> ### **References**
>
> [1] Safety alignment can be not superficial with explicit safety signals, Li et al. [ICML 2025]
>
> [2] Xie, Tinghao, et al. "SORRY-Bench: Systematically Evaluating Large Language Model Safety Refusal." [ICLR 2024]

---

### Author Response · Authors · 2025-11-19
**To all Reviewers and AC:  SSAH - Hypothesis Rather Than Formal Theorem**

Dear all reviewers and AC:

Our work is best understood as a **hypothesis-driven framework**, rather than a new defense **technique**. The core contribution is **SSAH** itself: we formally propose safety alignment as an implicit binary decision process and derive concrete corollaries (e.g., “less is more for safety alignment”). As we clearly state in the paper, **SSAH** cannot be fully proved in the current LLM setting (``lines 160-165, lines 215-217``); instead, we provide **multiple, complementary empirical signals**—probing results and unit-level analyses—that are **``consistent``** with its predictions and help **``explain known phenomena such as brittleness and alignment tax``**. We respectfully ask that the paper be **evaluated on its value as a conceptual and mechanistic lens for understanding safety alignment, not solely on whether it introduces a new solid advanced technique**.

Authors.

---

### Author Response · Authors · 2025-11-27
**Global update on the revised PDF: all changes are highlighted in blue.**

We have uploaded a revised version of the paper, with changes in blue. The changes are deliberate to address the concerns raised by reviewer **HeGB**.  Specifically, we add one section (``Appendix A``) to the appendix, which follows the following structure.


- Appendix A: More Related Works and Baselines
- Appendix A1: Overview of Safety-Unit and Safety-Alignment Work (18 papers)
- Appendix A2: Empirical Comparisons with Safety-Unit Freezing Baselines (2 new baselines)

We addressed all other points raised by the reviewers in detail in this rebuttal and will incorporate them in the camera-ready version if this paper is accepted.

---

### Author Response · Authors · 2025-12-01
**Summary for the Area Chair**

Dear AC:

To save your time, we briefly summarize our paper and how we addressed the concerns from the reviewers with **negative** scores, **HeGB**, and **asEz**.

**Our paper:**

* **SSAH as a safety explanation framework**. We propose SSAH, viewing safety alignment as an **implicit binary decision** (fulfill vs. refuse), and support it with probing analyses that reveal systematic shifts in *“reasoning direction”* between aligned and unaligned models.

* **Mechanistic analysis of brittleness and alignment tax**. Using the **SCU/UCU/CU/RU** taxonomy, we demonstrate that fine-tuning reassigns roles among these units (e.g., SCU→RU, CU→UCU), providing a concrete unit-level mechanism for both *safety brittleness* and the *alignment tax*.

* **“Less is more” as an implication of SSAH**. We show that safety behavior is governed by a small subset of units: freezing SCUs plus a small fraction of CUs with high **$I_{\text{safety}} - I_{\text{utility}}$** largely preserves safety, while the RUs can be fine-tuned as an explicit *“alignment budget”* to adapt utility with minimal alignment tax.

* **Toward a non-superficial SSAH.** We further outline a non-superficial extension of **SSAH**, where the model **re-evaluates** its safety decision at each decoding step, providing a conceptual path toward more robust defenses against *jailbreak-style attacks*.


**Regarding Review of HeGB**

* We had resolved all previous concerns, and then to resolve the reviewer’s last concern, we provided comparisons with **18 suggested related works** and implemented **two closest baselines** (safety-unit freezing methods), where our SCU+CU strategy consistently performs better under comparable or stricter parameter budgets.

**Regarding Review of asEz**

* The reviewer did not reply at all, but:

  1. Concern 1 was based on a misunderstanding—our probing setup already matches the suggested setting.
  2. Weakness 2 (about SSAH vs. “less is more”) is resolved by the similar clarifications (global & separate) that we gave to **HeGB** and by citing related work that builds on the binary-safety view (**SSAH**);
  3. Weaknesses 3 and 4 misunderstood that our main goal is to boost safety performance; instead, our work is a feasible explanation framework.

**Please let us know if you have further questions. Thank you again for your time and effort in this emergent situation.**

Author

---

### Meta-Review · Area_Chair_kzw1 · 2026-01-06

**Summary:**

This paper introduces the Superficial Safety Alignment Hypothesis (SSAH), which frames safety alignment in large language models as an implicit decision between fulfilling and refusing a request.
Reviewers agreed that the topic is important and that the paper contains extensive empirical analysis. Several reviewers found the hypothesis and the unit-level investigation novel and potentially useful as an explanatory lens. At the same time, concerns were raised about the conceptual grounding of SSAH, the logical relationship between the hypotheasis and its derived implications, and whether the empirical evidence sufficiently supports the central claims. In particular, one reviewer expressed strong reservations about deriving practical conclusions from an unverified hypothesis. Additional concerns included the initial lack of comparison to prior neuron-level safety methods.

**Reviewer Concerns:**

### Concerns addressed by the rebuttal:

The authors clarified that SSAH is intended as a hypothesis and explanatory framework, not a formally proven theory, and revised language that previously suggested stronger theoretical status.

Some missing related works were added.

Methodological details of the probing experiments were clarified, including how query semantics are handled, resolving concerns about confounding in the similarity measurements.

Reviewer HeGB's questions about the practical implications of the framework, alignment tax, and fine-tuning brittleness were addressed through additional experiments and discussion.

### Concerns that remain outstanding:

One reviewer remains unconvinced that the evidence sufficiently supports SSAH itself, rather than only its downstream implications, and continues to view the logical structure as speculative.

The connection between the binary-decision framing and the “less-is-more” findings is still questionable, particularly the reliance on implications derived from an unverified hypothesis.

**Reviewer Scores:**

Reviewer 4PEp: Likely to remain at 6 (weak accept) after clarifications.

Reviewer y2ma: Likely to remain at 6 (weak accept); concerns were largely addressed.

Reviewer asEz: Initially 4; after clarifications, may increse to 6.

Reviewer HeGB: initially 2, but the authors provided a very strong rebuttal, I think the rating may increase to 4 or 6.

---

### Decision · Program_Chairs · 2026-01-26

Accept (Poster)